ecology

invasions, demography, competition, phylogeny, functional traits

**Author for correspondence:**
Sam C. Levin
e-mail: levisc8@gmail.com

# Phylogenetic and functional distinctiveness explain alien plant population responses to competition

Sam C. Levin[1,2], Raelene M. Crandall[3], Tyler Pokoski[4], Claudia Stein[5,6] and Tiffany M. Knight[1,2,7]

[1]Martin Luther University Halle-Wittenberg, Institute of Geobotany, Am Kirchtor 1, 06108 Halle (Saale), Germany
[2]German Centre for Integrative Biodiversity (iDiv) Halle-Jena-Leipzig, Deutscher Platz 5e, 04103 Leipzig, Germany
[3]School of Forest Resources and Conservation, University of Florida, Gainesville, FL 32611, USA
[4]Department of Botany and Plant Pathology, Oregon State University, Corvallis, OR 97331, USA
[5]Department of Biology, Washington University of St Louis Tyson Research Center, 6750 Tyson Valley Road, Eureka, MO 63025, USA
[6]Department of Biology and Environmental Science, Auburn University at Montgomery, PO Box 244023, Montgomery, AL 36124-4023, USA
[7]Department of Community Ecology, Helmholtz Center for Environmental Research – UFZ, Theodor-Lieser-Straße 4, 06120 Halle (Saale), Germany

SCL, 0000-0002-3289-9925; RMC, 0000-0002-0229-5418; CS, 0000-0002-9586-8587; TMK, 0000-0003-0318-1567

Several invasion hypotheses predict a positive association between phylogenetic and functional distinctiveness of aliens and their performance, leading to the idea that distinct aliens compete less with their resident communities. However, synthetic pattern relationships between distinctiveness and alien performance and direct tests of competition as the driving mechanism have not been forthcoming. This is likely because different patterns are observed at different spatial grains, because functional trait and phylogenetic information are often incomplete, and because of the need for competition experiments that measure demographic responses across a variety of alien species that vary in their distinctiveness. We conduct a competitor removal experiment and parameterize matrix population and integral projection models for 14 alien plant species. More novel aliens compete less strongly with co-occurring species in their community, but these results dissipate at a larger spatial grain of investigation. Further, we find that functional traits used in conjunction with phylogeny improve our ability to explain competitive responses. Our investigation shows that competition is an important mechanism underlying the differential success of alien species.

## 1. Introduction

Alien species that have been transported outside of their native range by humans can sometimes dominate local communities and become invasive [1]. Such invasive species can often wreak havoc on ecological and economic systems [2,3]. Understanding why some alien species become so dominant in their new range, while others establish but fail to become dominant is necessary to forecast impacts and future invasions as well as to formulate mitigation strategies. However, a mechanistic understanding of the invasion process has been largely elusive [4–6].

A large number of hypotheses have been invoked to explain how some alien species come to invade and dominate their non-native communities, while other alien species are more benign [7,8]. Some of these involve properties of

the environment (e.g. disturbance, resource availability) and some involve properties of the species and their interactions with the resident community (e.g. competition, release from enemies). Here, we focus explicitly on how interactions with heterospecific plant species influence the population dynamics of alien plant species.

A prominent category of hypotheses invoked to explain invasiveness involve the strength of negative interspecific interactions with the resident community. This includes competition for limiting resources (e.g. water, nutrients) or mutualistic partners (e.g. mycorrhizae or pollinators), and indirect effects mediated through enemies (including pathogen and/or herbivores). For example, classic hypotheses going back to Darwin (1859) suggest that alien species which are less similar (evolutionarily and/or functionally) to the resident community will be less influenced by competition than those that are more similar. The idea here is that resident community should provide weak resistance to alien species that are functionally distinct, allowing these species to have high performance [9–12].

Despite its prominence in the literature, empirical support for the notion that more distinct species should have higher performance has been mixed [4,13,14]. There are several reasons for this lack of consensus, including: (i) coexistence and community assembly theory, which when applied to alien species suggests that species should perform better and be more likely to coexist if they are more different in certain ways, but more similar in other ways [15,16]; (ii) methodological differences across studies in the measures of distinctiveness [13,14]: phylogenetic and functional measures of distinctiveness provide complementary information (e.g. [17–21]), but are usually considered separately in the context of invasions (e.g. [22]); (iii) multiple types of data are often employed to test hypotheses—including presence-absence data, relative abundance data and performance data [14]—even though they can provide different answers and may relate to different stages of the invasion process; and (iv) the spatial grain in which the resident community is defined [13,23,24].

Experiments are a robust way to test mechanisms underlying patterns related to species interactions and alien performance. While several studies have experimentally tested whether plants compete more strongly with their close relatives, most of these have measured short-term responses of plant fitness in greenhouse experiments or propagule addition experiments (e.g. [25–28]). However, in the context of successful biological invasions, the effect of competition on a focal alien plant species is ideally measured in a field setting and considers lifetime fitness. Matrix projection models and integral projection models (MPMs and IPMs, respectively) are tools for summarizing fitness components measured in the field across the life cycle of plants and for comparing plant performance between experimental treatments [29–31]. To test whether distinct alien plant species compete less strongly with the resident plant community, it is necessary to quantify the population dynamics of multiple alien plant species that vary in their distinctiveness in treatments that manipulate the competitive environment.

We investigate the mechanisms underlying the differential success of alien plant species. We parameterize matrix projection and integral projection models for alien plant species in the presence and absence of competitors to ask whether phylogenetic distinctiveness predicts the strength of competitive interactions. We test whether the result found at a small spatial grain is robust even if the resident community is defined at a larger spatial grain that would be typical for studies using species checklists. Additionally, we quantify functional traits for 116 plant species and ask whether simultaneously incorporating information on functional traits and phylogenetic relationships improves the relationship between distinctiveness and the strength of competitive interactions, and, if so, which functional traits play key roles in explaining the effect size of competition.

We show that phylogenetically distinct species compete less with their local communities, but this relationship disappears if the resident community is defined at a larger spatial grain. Furthermore, incorporating functional traits provides additional explanatory power to our models, adding mechanistic support for competition underpinning differential success of alien plant species.

## 2. Methods

### (a) Study site and species

This study was conducted primarily at Washington University's Tyson Research Center (TRC), an 800 ha field station that is located 32 km southwest of St Louis, Missouri in the central United States (38°31′4.1″ N, 90°33′27″ W). Habitats within TRC include deciduous oak-hickory forests, prairies and Ozark glades. TRC is located close to suburban habitats and a major highway, and many alien plants have naturally established populations within the field station. In addition, one of the 14 plant species that was the focus of the competitor removal experiment was sampled from populations at the nearby Shaw Nature Reserve, which contains similar habitats to TRC. Our choice of 14 study species (table 1) for the competitor removal experiment was based on phylogeny (spread across the dicot tree of life, with some replication in large families) and opportunity (enough individuals for experimentation).

### (b) Experimental design

To experimentally assess the effects of resident competition on the performance of each alien plant species, we conducted a heterospecific competitor removal experiment. We established between 10 and 20 plots per focal species along transects of each population of focal species (though for some species, plots were lost resulting in fewer than 10 plots; see table 1 and [32]). The number of plots used to sample a plant species was determined by population densities and stage distributions, as it was necessary to sample individuals across a range of stages and sizes to parameterize the demographic models. When populations of focal species overlapped, separate plots were established so that the competitor removal treatment could be applied to each focal species separately.

Plots were randomly assigned into treatments: control (unmanipulated in any way) and competitor removal (figure 1a). For the competitor removal plots, we removed all individuals of non-focal species. The removal was carried out by clipping all biomass of non-focal species at ground level using scissors. This method ensured minimal soil disturbance, but also necessitated return trips (every 1 to 2 weeks, depending on the speed of regrowth in the different communities) to remove re-sprouting biomass.

The biomass of competitors varied across the populations of our 14 alien plants and is an important variable to account for when modelling the effect of competitor removal on plant performance (see below). To measure the biomass of competitors, all clipped biomass in the first clipping event was bagged, dried at 80°C for approximately 3 days and then weighed. We additionally removed all above ground biomass in an

**Table 1.** Information on each of the focal species, habitat they were studied in, plot size and type of population model used. (MPM, matrix projection model; IPM, integral projection model.)

| species | family | life history/ growth form | habitat type | plot size | plot number | MPM/ IPM |
|---|---|---|---|---|---|---|
| *Ailanthus altissima* | Simauroubaceae | tree | forest | 2 m × 2 m | 16 | IPM |
| *Alliaria petiolata* | Brassicaceae | monocarpic perennial | forest | 1 m × 1 mm | 9 | MPM |
| *Carduus nutans* | Asteraceae | monocarpic perennial | old field | 1 m × 1 m | 11 | MPM |
| *Draba verna* | Brassicaceae | winter annual | rocky outcropping | 0.5 m × 0.5 m | 10 | MPM |
| *Euonymus alatus* | Celastraceae | woody shrub | forest | 1 m × 1 m | 17 | IPM |
| *Kummerowia striata* | Fabaceae | summer annual | old field | 0.25 m × 0.25 m | 10 | MPM |
| *Lepidium campestre* | Brassicaceae | winter annual | old field | 1 m × 1 m | 8 | MPM |
| *Lespedeza cuneata* | Fabaceae | perennial herb | old field | 0.5 m × 0.5 m | 16 | MPM |
| *Ligustrum obtusifolium* | Oleaceae | woody shrub | forest | 1 m × 1 m | 10 | IPM |
| *Lonicera maackii* | Caprifoliaceae | woody shrub | forest | 2 m × 2 m | 10 | IPM |
| *Perilla frutescens* | Lamiaceae | summer annual | forest | 0.5 m × 0.5 m | 16 | MPM |
| *Potentilla recta* | Rosaceae | perennial herb | rocky outcropping | 0.5 m × 0.5 m | 12 | MPM |
| *Thlaspi perfoliatum* | Brassicaceae | winter annual | rocky outcropping | 0.5 m × 0.5 m | 7 | MPM |
| *Verbascum thapsus* | Scrophulariaceae | monocarpic perennial | rocky outcropping | 1 m × 1 m | 15 | MPM |

approximately 25 cm radius around the edges of each plot to prevent any edge effects, but this biomass was not included in our measurement of the biomass of competitors.

## (c) Alien plant performance and effect size of competitor removal

For each focal alien species, we marked individuals in the control and competitor removal plots and tracked them for two growing seasons. We collected information on survival, growth and reproduction (both sexual and asexual) to parameterize matrix population models or integral projection models (MPMs/IPMs) for each species and treatment. Some species' demography is best described by discrete stage, age or size classifications. MPMs are the most appropriate tool for modelling populations structured in this way [29]. Other species' demography, primarily trees, are best modelled with continuously distributed predictors (e.g. height, diameter at breast height, [30,31]). Individuals were pooled within each species and treatment combination to increase the sample size for parameter estimates. Information on seed bank transitions for species with dormancy was determined from the literature or from our own seed sowing experiments (see [32]). IPMs were discretized using the midpoint rule of integration and 500 meshpoints to generate projection matrices [31]. For each species and treatment, the population growth rate ($\lambda$) was calculated as the dominant eigenvalue of the projection matrix [29,31]. Models were re-fitted with bootstrapped datasets, and we computed $\lambda$ and the effect size of competition during each iteration of the procedure to generate 1000 estimates of each (figure 1b). This allowed us to incorporate uncertainty in the demographic data into the models we describe below. The comparison of results from MPMs and IPMs has precedence in the comparative demographic literature and has not been found to confound results (e.g. [33]). The demographic data and further species-specific details for demographic data collection and model construction is available as a data paper [32].

The effect size of competitor removal on $\lambda$ for each species was calculated as the log response ratio (equation (2.1)):

$$\text{effect size of competition}_i = \frac{\ln(\lambda_{i,CR} + 0.5)}{\ln(\lambda_{i,C} + 0.5)}. \quad (2.1)$$

For species $i$, $\lambda_{i,CR}$ is the population growth rate of species $i$ in the competitor removal treatment, and $\lambda_{i,C}$ is the population growth rate of species $i$ in the control treatment ($y$-axis of figure 1c). We add 0.5 in both the numerator and denominator to adjust for lambdas close to zero that occasionally occur in the control treatment [34].

## (d) Resident plant community at two spatial grains

To determine the identity and abundance of co-occurring plant species at the small spatial grain, we sampled the community at the same time as our first demographic sampling period for each focal alien species. In each control plot, we documented the identity and the per cent cover of all species. Grasses were identified to the genus level. Species that were clearly important to the community, but not necessarily present in plots (e.g. canopy tree species at forest understory sites) were also identified and included in the community for the small spatial grain analyses, but no abundances were recorded (electronic supplementary material, Appendix S1). The identity of co-occurring plant species at the larger spatial grain was based on the checklist of plants for the 800 ha TRC field station.

## (e) Phylogenetic distinctiveness

To create a measure of phylogenetic distinctiveness for each alien plant species at each spatial grain, we first used a time-calibrated phylogenetic tree of angiosperms [35] and inserted species present at the large spatial grain (both native and alien) that were not already present in this tree by creating a polytomy with congeners using the *congeneric.merge* function from the *pez* package for R [36]. The phylogeny was then pruned to only include species present at Tyson; this is the TRC tree. This Tyson phylogeny contains all species in our regional species pool.

Distinctiveness at the large spatial grain was calculated using two metrics, mean pairwise distance (MPD) and nearest neighbour distance (NND). A phylogenetic distance matrix (between all species in the TRC tree), MPD and NND were calculated using the *cophenetic.phylo* method from the *ape* R package [37]. The distance matrix was square root transformed to facilitate comparisons to and combination with functional trait information [38].

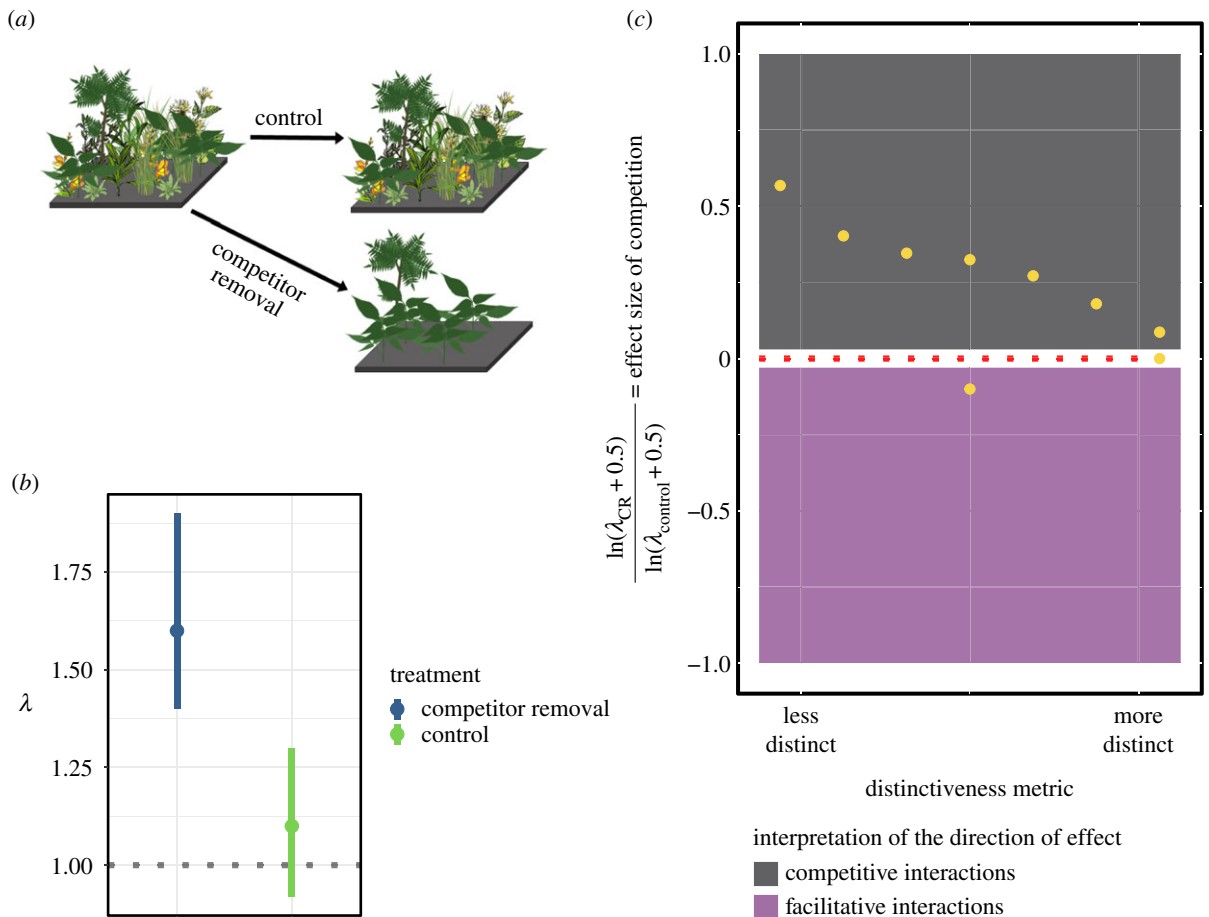

**Figure 1.** A conceptual overview of the field experiment, demographic effect size computations and interpretation of the regression analyses. (*a*) depicts the methods of the competitor removal experiment, where all individuals of non-focal species were removed from half of the plots (lower portion of panel). The control plots were not manipulated in any way (upper portion of panel). (*b*) depicts the hypothetical distributions of the population growth rate ($\lambda$) of two focal exotic species in each treatment. (*c*) graphically depicts a single iteration of the distinctiveness regressions. Each point on the figure represents the effect size of competition for a single species (e.g. log response ratio of competitor removal and control treatments). Points above 0 indicate that removing competitors significantly increased $\lambda$ compared to the control treatment, while points below 0 indicates removing competitors significantly decreased $\lambda$ compared to the control treatment (expected if non-focal species facilitate rather than compete with focal aliens). The dotted line at 0 indicates the case where there was no effect of the competitor removal treatment on $\lambda$. Yellow points are hypothetical log response ratios for nine species. These iterations were repeated 1000 times, from the bootstrapping procedure. The distributions of regression coefficients for each distinctiveness metric are shown in figure 2. (Online version in colour.)

Distinctiveness at the small spatial grain was calculated using four metrics: MPD and NND, as well as abundance weighted versions of MPD and NND (termed AW-MPD and AW-NND, respectively). For each alien species, the phylogeny was pruned to create 14 community-scale phylogenies containing the focal species and species they co-occur with in the demography control plots (community-level tree). We calculated a phylogenetic distance matrix between all species in each community-level tree. To allow for a species richness-standardized comparison of distinctiveness across all 14 focal alien species, NND was calculated as the average distance to nearest neighbour of 1000 richness rarefied samplings. To rarefy, we randomly chose 11 species from each small grain community and re-built the community-scale phylogeny. We used 11 species because this was the number of co-occurring species present in the least species-rich community (focal alien: *Ligustrum obtusifolium*). We also rarefied MPD for each of the 14 alien species, even though MPD is known to be less sensitive to species richness than NND [39]. To calculate AW-MPD and AW-NND, we wrote a thin wrapper around the *mpd* function in *picante* to accommodate the structure of our data ([29], available in *FunPhylo* package, https://github.com/levisc8/Fun_Phylo_Package).

To quantify how our results changed as a function of phylogeny construction method, we obtained two other large

angiosperm phylogenies and re-ran all analyses using them [40]. Briefly, one phylogeny uses all available sequences on GenBank and generates a phylogeny for those using the Open Tree of Life backbone [40] while the other phylogeny uses both the GenBank data and other taxonomic information to generate a far more comprehensive, though less reliable phylogeny using the same Open Tree of Life backbone [40]. The results did not change when these were substituted in (see results in the electronic supplementary material, Appendix S2).

## (f) Relationship between phylogenetic distinctiveness and effect size of competitor removal

To test the relationship between phylogenetic distinctiveness and the effect size of competitor removal, we constructed linear models using metrics of phylogenetic distinctiveness and competitor removal biomass as explanatory variables and the effect size of competitor removal on $\lambda$ (equation 2.1) as the response variable ($N = 14$ focal species). Phylogenetic distinctiveness was calculated at small (plot) and large (TRC) grains. Separate linear models were created for each metric of phylogenetic distinctiveness and each spatial grain. For the small grain we rarified (see above) and log transformed the abundance weighted

phylogenetic distinctiveness metrics prior to analysis. To test whether uncertainty in the demographic data affected our overall conclusions, we re-fitted the models described above 1000 times using the effect size of competition from the demographic boot-strapping procedure and stored the coefficients and $R^2_{\text{adj}}$ from every iteration.

*Kummerowia striata* had an exceptionally large effect size of competition and co-occurred with a congener, *Kummerowia stipulacea*, making it a notable outlier. We re-ran all analyses without this focal species and found that our main conclusions are robust to this outlier (electronic supplementary material, Appendix S2).

## (g) Functional traits

We collected data on plant functional traits for 116 of the 555 dicot species at TRC. We focused on species that were present in our plots and/or were close relatives of the 14 focal alien species at the regional spatial grain. We measured plant height, specific leaf area (SLA) and leaf toughness using standardized methods ([41], see the electronic supplementary material, Appendix S3 for details). We sampled at least 10 individuals per species. We conducted literature searches to include information on growth form, ability to fix nitrogen, month of first flowering (circular variable, [42]), dispersal syndrome and ability to reproduce clonally. For communities surrounding the woody focal species, we also include information on wood density from the *BIOMASS* package [43] and data collected by other researchers at the TRC field station [44]. In total, we collected a species-level value for each trait for greater than 75% individuals in each plot (and wood density for at least approximately 49% of individuals of woody species). Results for trait coverage by trait and/or habitat type are show in the electronic supplementary material, Appendix S3 (table S3.4 and figure S3.1).

## (h) Functional trait analyses

We also tested whether incorporating functional trait distinctiveness into regressions of phylogenetic distinctiveness improved the overall models. We hypothesized that traits which are not phylogenetically conserved would be most informative when combined with phylogenetic information. Pairwise continuous trait correlations were tested using Pearson correlation coefficients and highly correlated traits were removed from our analysis to avoid including redundant information (electronic supplementary material, Appendix S3).

We tested for phylogenetic signal in the trait data *Blomberg's K* for continuous traits [45,46] and the *D* statistic for binary traits (our categorical traits, growth form and dispersal method, were split into dummy variables [47,48]). We calculated a distance matrix for the circular variable, month of first flowering, and then performed a Mantel test with this distance matrix and a square root transformed phylogenetic distance matrix to test for phylogenetic signal [38,42,49]. We repeated this procedure using all traits to create the distance matrix to check for multivariate trait signal in phylogeny. The last two procedures were implemented using the *ade4* [49,50] and *vegan* [51] packages for R. Wrappers for these analyses are available in the *FunPhylo* package.

Owing to our sample size of 14 species, it was not possible to keep each trait plus phylogenetic distinctiveness as separate predictors in a model of competitive responses. Thus, we used an integrated distance metric derived by Cadotte and colleagues [18] to determine how distinctiveness from the resident community predicted an alien's competitive response (see also the interactive web app we created using the *shiny* R package [52]: https://sam-levin.shinyapps.io/Invasives_FPD/). We calculated functional trait distance matrices for each community containing our 14 focal alien species using a modified version of the Gower distance introduced by Pavoine *et al.* [42] to include continuous variables (SLA, height, toughness, wood density), circular

variables (month of first flowering) and binary variables (growth form, dispersal mechanism). Next, we used a modelling approach that varies the weight (*a*) that phylogenetic distinctiveness was given in calculating the distinctiveness values ([18], equation (2.2)):

$$FPD = \sqrt{a * (\text{phylogenetic novelty})^2 + (1 - a) * (\text{functional novelty})^2}.$$

(2.2)

Here, FPD is the combined functional-phylogenetic distinctiveness where *a* values of 1 correspond to only phylogenetic information, *a* values of 0 correspond to only functional trait information and values in between indicate intermediate weighting of each. We varied *a* in increments of 0.025 and extracted MPD, AW-MPD, NND and AW-NND for each of our focal species. Next, we regressed the observed values of effect size of competition on those distinctiveness values and the standardized competitor biomass and extracted the models' $R^2_{\text{adj}}$. The set of traits and *a* values that provided the best fit for the data are presented here, and the app as well as the code and custom R package to power it are publicly available (https://github.com/levisc8/Fun_Phylo_Shiny, https://github.com/levisc8/Fun_Phylo_Package). We did test the explanatory power of traits alone (i.e. when *a* = 0), but these models were never better at explaining competitive effect size as phylogeny-only or the combined functional-phylogenetic metric. These results are shown in the electronic supplementary material, Appendix S3 (figures S3.2–11). All analyses were conducted in R v. 4.0.0 [53].

# 3. Results

## (a) Demography experiment results

The effect size of the competitor removal treatment on $\lambda$ had a consistently negative relationship with phylogenetic distinctiveness at small spatial grains, but not at large spatial grains (figure 2). MPD explained more variance in the relationship between the effect size of the competitor removal and phylogenetic distinctiveness at small spatial grains than NND in both unweighted and abundance weighted analyses (figure 2 and table 2).

## (b) Phylogenetic signal in functional traits

The results of the functional trait analysis for continuous traits indicated that SLA, plant height and wood density were phylogenetically conserved, while leaf toughness was not (electronic supplementary material, table S3.1). Our categorical traits exhibited varying degrees of phylogenetic signal, with some dispersal traits and one growth form displaying conservatism (dispersal: unassisted, wind, ant dispersed and water dispersed, growth: vine) under a Brownian motion model ($D \leq 0$), while all other growth forms and dispersal syndromes showed no pattern or significant overdispersion ($D \geq 1$) (electronic supplementary material, table S3.1). The Mantel test for first flower time indicated that it was not correlated with phylogeny in our dataset ($r = 0.0174$, $p = 0.299$). The Mantel test using phylogenetic distance matrix and the distance matrix with all functional traits showed that the combined suite of traits weakly covaried with phylogeny ($r = 0.2166$, $p = 0.001$).

## (c) Functional-phylogenetic model performance

Our modelling approach using functional and phylogenetic information indicated that phylogeny alone explains much of the variance in the effect size of competitor removal on

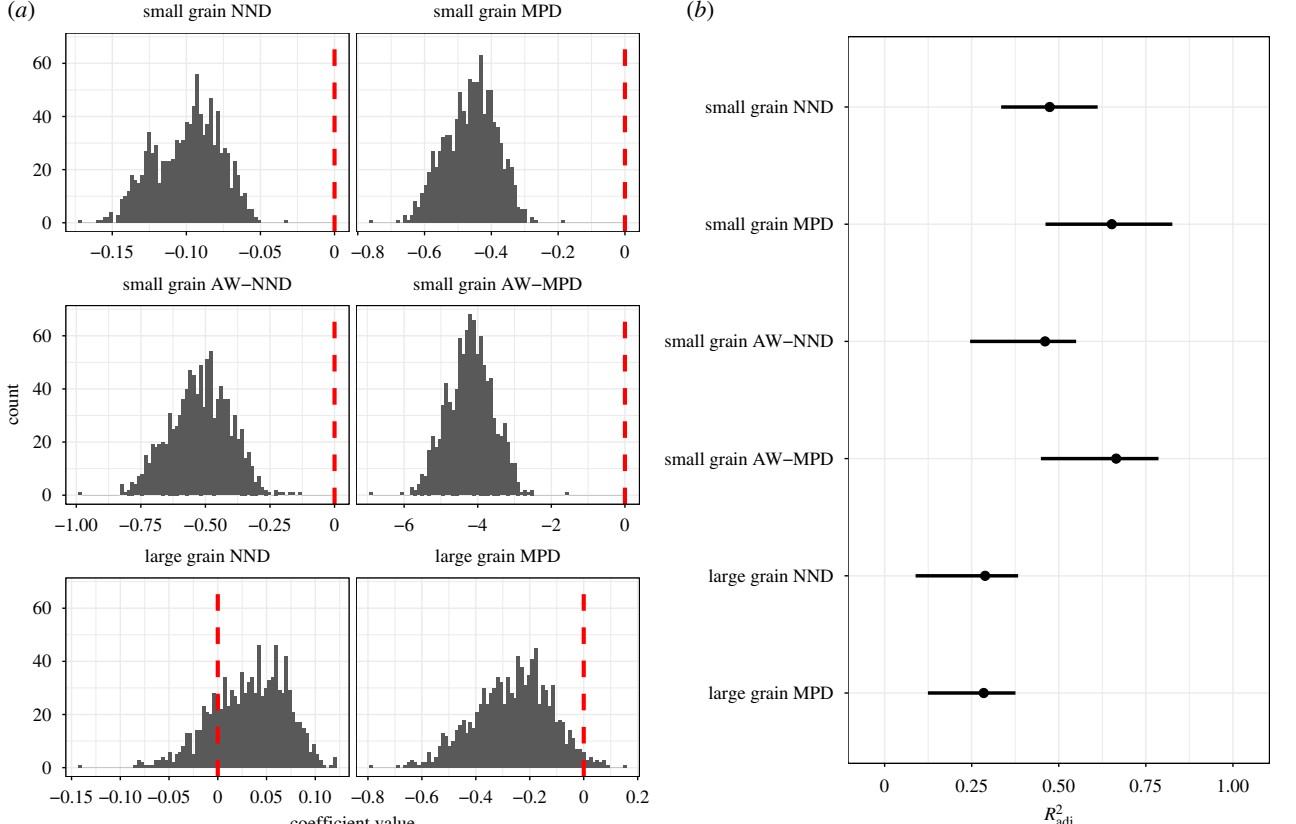

**Figure 2.** (a) Histograms of the regression coefficients for the relationship between the effect size of competitor removal and distinctiveness for 1000 bootstrap iterations. Histograms show each distinctiveness metric and spatial grain. Red vertical lines are placed at 0, indicating no relationship between the effect size of competitor removal and distinctiveness. Negative values indicate that the effect size of competitors removal decreases with distinctiveness. (b) Points showing the observed values and 95% confidence intervals for $R^2_{adj}$ for 1000 bootstrap iterations for each distinctiveness metric and spatial grain. MPD, mean pairwise distance; NND, nearest neighbour distance; AW-MPD, abundance weighted mean pairwise distance; AW-NND, abundance weighted nearest neighbour distance. (Online version in colour.)

**Table 2.** Phylogenetic distinctiveness coefficients and their bootstrapped confidence intervals (CI) for phylogeny-only models. (CI are upper and lower 95% CI.)

| spatial grain | parameter | observed value | lower CI | upper CI |
|---|---|---|---|---|
| small | MPD | −0.42760 | −0.60732 | −0.32910 |
|  | $R^2_{adj}$ | 0.65227 | 0.46196 | 0.82604 |
| small | NND | −0.08817 | −0.14207 | −0.06211 |
|  | $R^2_{adj}$ | 0.47393 | 0.33463 | 0.61174 |
| small | AW-MPD | −4.02658 | −5.38598 | −3.04082 |
|  | $R^2_{adj}$ | 0.66481 | 0.44874 | 0.78632 |
| small | AW-NND | −0.49514 | −0.74934 | −0.31271 |
|  | $R^2_{adj}$ | 0.46069 | 0.24550 | 0.54983 |
| large | MPD | −0.18288 | −0.54939 | −0.00801 |
|  | $R^2_{adj}$ | 0.28464 | 0.12429 | 0.37574 |
| large | NND | 0.05653 | −0.05064 | 0.09484 |
|  | $R^2_{adj}$ | 0.28857 | 0.08889 | 0.38329 |

plant population growth rates, and functional traits did improve models when weighted at an intermediate level. This pattern depends on choice of traits and distinctiveness metric used. Incorporating information on SLA, plant height, leaf toughness and month of first flowering, and using NND as the distinctiveness metric yielded the highest possible explanatory power for these data (figure 3; NND, maximum $R^2_{adj} = 0.784$, $a = 0.175$).

## 4. Discussion

### (a) Mechanisms are scale dependent

This study provides experimental evidence that competition is a key mechanism underlying patterns of distinctiveness and alien plant performance, and that patterns depend on spatial grain. There was a consistent negative relationship between distinctiveness and the effect size of competition

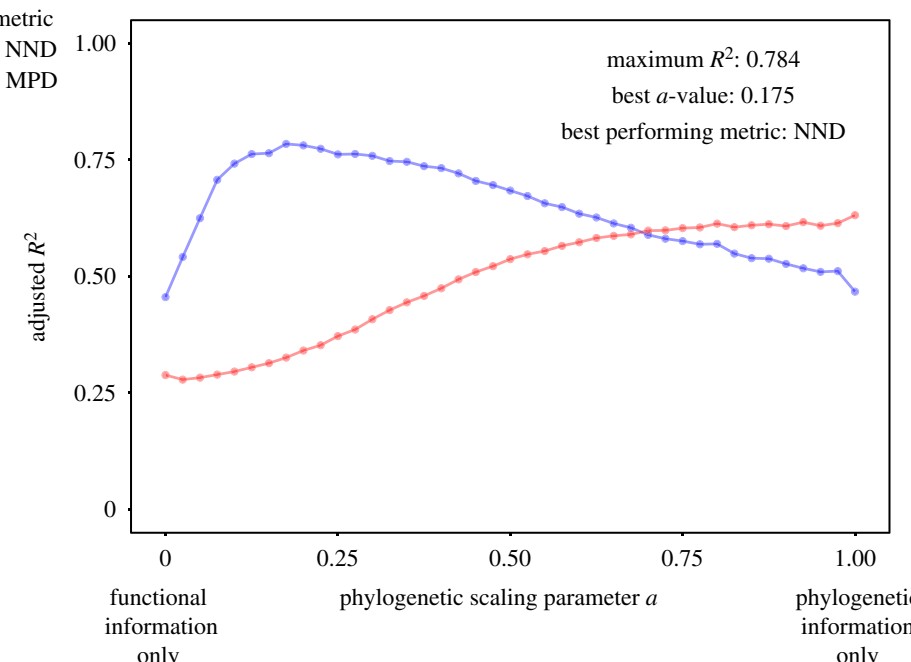

**Figure 3.** Explanatory power of functional-phylogenetic regression models as a function of the phylogenetic scaling parameter, *a*. Values of 0 indicate only functional trait information while values of 1 indicate only phylogenetic information. Intermediate values indicate varying degrees of weighting given to each type of information when calculating distinctiveness. (Online version in colour.)

(figure 2). We expected this relationship would be stronger at small spatial grains in which plants compete than at larger grains. Indeed, the slope of the relationship is not distinguishable from 0 when the resident species are defined at a larger spatial grain (figure 2). Recent work has highlighted the pitfalls of using co-occurrence as a proxy for interactions, and our findings provide further support for this [54,55]. Our direct manipulation of competition and quantification of demographic performance confirms hypothesized relationships between spatial grain, distinctiveness and competition between invasive and resident species [23,25,56–63] and does so in a natural setting.

## (b) Functional-phylogenetic models

Models using only phylogenetic information performed better than models using only functional trait information, but models containing both types of information performed best in explaining the variance in responses to competitive interactions (figure 3). We found that differences in phylogenetically conserved traits specifically related to competitive ability and growth rate (SLA, height, [41,64]), reproductive capacity (month of first flowering) and leaf defence (leaf toughness, [41]) were most useful in improving model fits. These models, parameterized with data from an experimental field setting, also provide a mechanistic link between competition and the traits that give rise to the differential successes of alien plant species.

Contrary to our original hypothesis that unconserved traits would provide the most useful information, we found that a mixture of phylogenetically conserved and un-conserved traits displayed the most predictive power. There are two non-mutually exclusive explanations for this: (i) the conserved traits are probably more important determinants of competitive outcomes, and (ii) there is still variation across species in conserved traits that is not explained by phylogeny [19]. This was particularly true for SLA (electronic supplementary

material, table S3.1) and is perhaps an indicator as to why this combination of traits in conjunction with phylogeny could perform better than the phylogeny-only models.

## (c) Relationship to invasion stage

In this study, we consider a single stage in the invasion process: whether or not already established alien plant species become dominant. We do not consider whether phylogeny and traits determine which alien species are able to successfully establish in our region. In our case, this is not possible owing to a lack of information on failed introductions. Experiments manipulating the relatedness of the invader to the invaded community have found that relatedness to the community predicts establishment success for microbes [65]. Older work found this was not true for vascular plants [66], but more recent experimental work has found nonlinear relationships between relatedness and establishment success [63]. Our experiment does provide robust evidence that competition mediates the relationship between distinctiveness and subsequent dominance of alien plant species.

At any particular site, mechanisms that explain alien success are likely to change with habitat succession and/or increased duration of invasion. Indeed, traits best explaining rate of establishment and subsequent spread have been found to differ, supporting the idea that multiple mechanisms act throughout the course of an invasion to determine its long-term success [55]. In our case, we considered a single population of each focal alien species and did not consider how spatial and temporal variation in the successional stage of the habitat influenced plant demography and its response to competitor manipulations.

## 5. Conclusion

We show through experimental manipulation that competition mediates the relationship between phylogenetic

distinctiveness and invasiveness at small, but not large spatial grains. Incorporating a few easily measured functional traits enhanced our ability to predict the strength of competitive dynamics, though phylogeny alone was fairly successful. Our investigation shows that competition is an important mechanism underlying the differential success of invasions following the establishment phase.

Data accessibility. The raw demographic data and code to build the matrix and integral projection models are available freely at https://doi.org/10.5281/zenodo.2573061. The only condition for use is to cite Levin et al. ([32] in this manuscript). The code and data to reproduce the phylogenetic and functional trait analyses are available freely at https://github.com/levisc8/Thesis_SL. This code depends on the FunPhylo R package and the data stored therein. This is available at https://github.com/levisc8/Fun_Phylo_Package.

Competing interests. We declare we have no competing interests.

Funding. This study was funded by the National Science Foundation (DEB 1145274), the Helmholtz Association in the framework of the Helmholtz Recruitment Initiative and the Alexander von Humboldt Foundation in the framework of the Alexander von Humboldt Professorship of T.M.K.

Acknowledgements. The data collection would not have been possible without Amibeth Thompson, Cassandra Galluppi, Erynn Maynard, Eleanor Pearson and the Tyson Plant Invaders crew. Support for the high school students on the Plant Invaders crew was provided by a grant to Washington University in St Louis from the National Science Foundation Advancing Informal STEM Education Program (DRL 0739874).

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
