## [Reviewer comments · Proceedings of the Royal Society B: Biological Sciences]

Review History

RSPB-2019-1695.R0 (Original submission)

Review form: Reviewer 1

Recommendation

Major revision is needed (please make suggestions in comments)

Scientific importance: Is the manuscript an original and important contribution to its field?

Good

General interest: Is the paper of sufficient general interest?

Good

Quality of the paper: Is the overall quality of the paper suitable?

Good

Is the length of the paper justified?

Yes

Should the paper be seen by a specialist statistical reviewer?

Yes

Do you have any concerns about statistical analyses in this paper? If so, please specify them explicitly in your report.

No

It is a condition of publication that authors make their supporting data, code and materials available - either as supplementary material or hosted in an external repository. Please rate, if applicable, the supporting data on the following criteria.

Is it accessible?

Yes

Is it clear?

No

Is it adequate?

Yes

Do you have any ethical concerns with this paper?

No

Comments to the Author

There have been various test of Darwin's Naturalisation Hypothesis but few have taken an experimental approach as in this study and thus this work does merit attention. Nevertheless, the authors made the the manuscript quite hard going. It was hard to navigate the manuscript without details of the models and calculation of lambda. I checked out the online supplement but feel that this still made it hard work to see exactly how the models were put together and lambda calculated. My assumption is that lambda was calculated as the number of individuals in the plot at time t+1 compared to time t but some of the plot sizes, especially for woody species, would be too small to calculate such a measure. The authors therefore need to provide a section on the demographic modelling approach and how the models were parameterized and how aspects like seedbank and fecundity were estimated by the authors. Without this detail it is hard to see exactly what was done and the robustness of their conclusions. Obviously this lengthens the manuscript and if material needs to be cut then perhaps it should related to the scaling issue, which is well known by now and perhaps is the less informative aspect in this manuscript.

Minor comments:

I would suggest the authors use the standard terms alien, invasive and naturalized and not confuse matters by using exotic in a way that has heretofore not been used.

Review form: Reviewer 2

Recommendation

Major revision is needed (please make suggestions in comments)

Scientific importance: Is the manuscript an original and important contribution to its field?

Good

General interest: Is the paper of sufficient general interest?

Good

Quality of the paper: Is the overall quality of the paper suitable?

Marginal

Is the length of the paper justified?

Yes

Should the paper be seen by a specialist statistical reviewer?

No

Do you have any concerns about statistical analyses in this paper? If so, please specify them explicitly in your report.

Yes

It is a condition of publication that authors make their supporting data, code and materials available - either as supplementary material or hosted in an external repository. Please rate, if applicable, the supporting data on the following criteria.

Is it accessible?

Yes

Is it clear?

N/A

Is it adequate?

N/A

Do you have any ethical concerns with this paper?

No

Comments to the Author

This is a potentially interesting study assessing the effect of removal of competition on the spread of alien species within a region. The authors have sampled vegetation in different plots (~10-20), across different habitats, for each of the 14 alien species selected. They removed all competitors in this plots and assessed changes in demographic parameters as a measure of the effect of competition on alien species (but, to my understanding, this was done only over one growing season). The effect of competition was significant on ~50% of the 14 species considered, generally only on invasive species, i.e. those which historically have been expanding more in the region (the authors distinguish invasive vs. exotic, where exotic have expanded less and only to disturbed conditions; i.e. invasive + exotic = alien, according to the terminology used in this study). The effect of competition was also stronger when the aliens were more phylogenetically similar species to the set of species removed (notice that here I use "species removed" and I haven't said "native", because the species removed were not always native; this aspect was not very clearly taken into account in the study). Measures of functional dissimilarity, based on some on-site trait measurements + bibliographic information, were also considered. However, functional dissimilarity alone was, if I understand well the results, not a good predictor when used alone (although I would like to see Fig. 1 and Table 2, also for different set of traits). Functional traits, at the same time, seem to improve some predictions based only on phylogenetic distance between species (based on the method proposed by Cadotte et al. 2013). Two measures of phylogenetic and functional diversity were considered, MPD and MNTD, and only on the second the combination of functional and phylogenetic information was relevant.

I have some major problems with study, in its present form:

1. Of course, as the authors already partially recognize, the study focus "only" on the effect of competition on the spreading of alien species, when they have already established. The results should be better put into a proper context. Also, removal experiment generally focus on the

removal of 1 or 2 dominant species, while here the authors remove all species, which is more a simulation of disturbance, to me, than competition alone. See also next point.

2. Following point 1, my hypothesis (before looking at the results) was that “competition” effects would be stronger on exotic species, i.e. those that are assumed to spread less because they are found only on disturbed sites (LL166-170). Hence the results could be seen also as a bit counterintuitive, i.e. the effect of competition was actually stronger on species that actually were already able to win the competition with native species, i.e. they have already expanded in the region. Clear hypotheses on the effect of competition on “invasive” vs. “exotic” species should be better introduced and discussed, particularly with the type of experiment considered.

3. Clearly the design is not well balanced in terms of phylogenetic distances considered. When compared to species removed, the values of phylogenetic distance of invasive species (Fig. 1) is likely smaller and more variable than the ones of exotic species. Is there a similar effect on trait dissimilarity? please show us. This seems to be a serious issue, for both MPD and MNTD. This problem cannot be solved now, but anyway the authors say in the methods that they tried to balance the selection of some invasive and exotic species within families. Where are these comparisons? Not in the results section, at least. These results are potentially important because they can validate, or not, the general results from this study. BTW, most of the results seem to be affected by one or two alien species, one being a tree, which was quite unique phylogenetically (and functionally). How much this/se species influence the results? Should we trust the generality of the results?

4. The description of methods is certainly not very clear, and need clear improvements. I had to spend a lot of time to try understanding what the authors did, and I am still not sure I got everything. Obviously this should not be the case. My general feeling is, anyway, that some decisions were not 100% clear and/or trustable, at least how they were explained now. I might surely have lost something, but I am afraid that the methods are not clear enough and they need to be justified and clarified. For example, authors say that they have 14 species and 10-20 plots for each. But then they say they sampled 108 vegetative plots. Is this incongruence referring to the issue that plots were (moreover) not fully independent spatially? BTW, this is an issue not well taken into account. Hence, the counting (108) is unclear. “Vegetative” is also unclear. Then they say “We removed all individuals of non-focal species” but later they say they have control plots. Introduce better these control plots, please, they come to readers’ surprise. How many plots do they have for each species? Add this into Table 1. A rarefaction for phylogenetic distance indices was done (randomly selecting 11 species in each “community” 1000 times; btw here I do not know also if community refers to the ~10 plots used for each alien species). I do not understand why simple null-models were not used, just shuffling the identity of the all species removed, out of all plots sampled for a species. And how was done this rarefaction at the regional level? Using also species from different habitats? Also I am not sure why authors need to create 14 community-scale phylogenies, the function ‘mpd’ and ‘mntd’ can use a bigger distance matrix than the single plots considered. I am thus a bit scared by the sentence “In this case, branch lengths were rescaled”, because different scaling in the 14 species would actually impede any type of comparison between species. What was this rescaling? Why it was needed? What is this modified mpd function doing? All this is unclear, and a bit scary. L302: how was the distance matrix on the circular trait considered (the method by Pavoine et al. 2009 was applied?)? What is the modified version of the Gower distance (L314) doing?

5. I understand that demographic parameters were established only during 1 growing season. Is it so? Is this not this a big problem for the estimation of the demographic models, which should take into account multiple growing seasons? Is it safe to use different types of population model (from Table 1)?

6. The statistical model described starting at line 283, is not very clear, please justify all decisions. I would like to see both results across species (like in the present Fig. 1) but also within species. So, it is not clear if the models in Fig. 1 and Fig. 2 are based on number of observation equal to 14

or to 108. I guess the second is true but, if so, then this means that the authors could present also results of test within species (with some sort of caterpillar plots or so). I understand there is a sort of gradient within the sampling of each species. Anyway a couple of decisions in the statistical test were a bit unclear/obscure. The 14 species are clearly a random selection in the region, hence the absence of species identity (of the alien species) as random factor seems arguable. Also it is not clear why the same model was not also considered for different combination of traits (or single traits), so that Fig. 1 and Table 1 can include also traits as well. Please clarify and show analyses in the main paper.

I finally invite the authors to consider a couple of recent papers:

By Jon Bennett: <https://onlinelibrary.wiley.com/doi/abs/10.1111/jvs.12779>

By Luisa Conti et al.: <https://besjournals.onlinelibrary.wiley.com/doi/abs/10.1111/1365-2745.12928>

Decision letter (RSPB-2019-1695.R0)

15-Aug-2019

Dear Mr Levin:

I am writing to inform you that your manuscript RSPB-2019-1695 entitled "Phylogenetic and functional novelty explain alien plant population responses to competition" has, in its current form, been rejected for publication in Proceedings B.

This action has been taken on the advice of referees, who have recommended that substantial revisions are necessary. With this in mind we would be happy to consider a resubmission, provided the comments of the referees are fully addressed. However please note that this is not a provisional acceptance.

Sincerely,

Professor Hans Heesterbeek

Associate Editor

Board Member: 1

Comments to Author:

We now have two reviews of this paper on alien plant population responses to competition. Both reviewer--and i agree--found this to be an interesting experimental addition to a mostly observational literature. There is certainly something here to offer. However, both reviewers also found the introduction to be clear but the methods and results to be very confusing. I agree. I liked the shinyapp, but the fact the complexity of the results needs this level of interactivity is and will be confusing for readers. Furthermore, the connection back to theory gets very hazy. The reviewers both have suggestions for clarifying specific parts, but in my view this requires a more substantial re-thinking about which are the important results here, and then streamlining the paper to clearly build to those results.

One detailed comment: square-root of phylogenetic distance rather than the raw values are a much better match for reasonable models of functional difference between species for reasons laid out in this paper:

Letten, Andrew D., and William K. Cornwell. "Trees, branches and (square) roots: why evolutionary relatedness is not linearly related to functional distance." *Methods in Ecology and Evolution* 6.4 (2015): 439-444.

Reviewer(s)' Comments to Author:

Referee: 1

Comments to the Author(s)

There have been various test of Darwin's Naturalisation Hypothesis but few have taken an experimental approach as in this study and thus this work does merit attention. Nevertheless, the authors made the the manuscript quite hard going. It was hard to navigate the manuscript without details of the models and calculation of lambda. I checked out the online supplement but feel that this still made it hard work to see exactly how the models were put together and lambda calculated. My assumption is that lambda was calculated as the number of individuals in the plot at time t+1 compared to time t but some of the plot sizes, especially for woody species, would be too small to calculate such a measure. The authors therefore need to provide a section on the demographic modelling approach and how the models were parameterized and how aspects like seedbank and fecundity were estimated by the authors. Without this detail it is hard to see exactly what was done and the robustness of their conclusions. Obviously this lengthens the manuscript and if material needs to be cut then perhaps it should related to the scaling issue, which is well known by now and perhaps is the less informative aspect in this manuscript.

Minor comments:

I would suggest the authors use the standard terms alien, invasive and naturalized and not confuse matters by using exotic in a way that has heretofore not been used.

Referee: 2

Comments to the Author(s)

This is a potentially interesting study assessing the effect of removal of competition on the spread of alien species within a region. The authors have sampled vegetation in different plots (~10-20), across different habitats, for each of the 14 alien species selected. They removed all competitors in this plots and assessed changes in demographic parameters as a measure of the effect of competition on alien species (but, to my understanding, this was done only over one growing season). The effect of competition was significant on ~50% of the 14 species considered, generally

only on invasive species, i.e. those which historically have been expanding more in the region (the authors distinguish invasive vs. exotic, where exotic have expanded less and only to disturbed conditions; i.e. invasive + exotic = alien, according to the terminology used in this study). The effect of competition was also stronger when the aliens were more phylogenetically similar species to the set of species removed (notice that here I use “species removed” and I haven’t said “native”, because the species removed were not always native; this aspect was not very clearly taken into account in the study). Measures of functional dissimilarity, based on some on-site trait measurements + bibliographic information, were also considered. However, functional dissimilarity alone was, if I understand well the results, not a good predictor when used alone (although I would like to see Fig. 1 and Table 2, also for different set of traits). Functional traits, at the same time, seem to improve some predictions based only on phylogenetic distance between species (based on the method proposed by Cadotte et al. 2013). Two measures of phylogenetic and functional diversity were considered, MPD and MNTD, and only on the second the combination of functional and phylogenetic information was relevant.

I have some major problems with study, in its present form:

1. Of course, as the authors already partially recognize, the study focus “only” on the effect of competition on the spreading of alien species, when they have already established. The results should be better put into a proper context. Also, removal experiment generally focus on the removal of 1 or 2 dominant species, while here the authors remove all species, which is more a simulation of disturbance, to me, than competition alone. See also next point.
2. Following point 1, my hypothesis (before looking at the results) was that “competition” effects would be stronger on exotic species, i.e. those that are assumed to spread less because they are found only on disturbed sites (LL166-170). Hence the results could be seen also as a bit counterintuitive, i.e. the effect of competition was actually stronger on species that actually were already able to win the competition with native species, i.e. they have already expanded in the region. Clear hypotheses on the effect of competition on “invasive” vs. “exotic” species should be better introduced and discussed, particularly with the type of experiment considered.
3. Clearly the design is not well balanced in terms of phylogenetic distances considered. When compared to species removed, the values of phylogenetic distance of invasive species (Fig. 1) is likely smaller and more variable than the ones of exotic species. Is there a similar effect on trait dissimilarity? please show us. This seems to be a serious issue, for both MPD and MNTD. This problem cannot be solved now, but anyway the authors say in the methods that they tried to balance the selection of some invasive and exotic species within families. Where are these comparisons? Not in the results section, at least. These results are potentially important because they can validate, or not, the general results from this study. BTW, most of the results seem to be affected by one or two alien species, one being a tree, which was quite unique phylogenetically (and functionally). How much this/se species influence the results? Should we trust the generality of the results?
4. The description of methods is certainly not very clear, and need clear improvements. I had to spend a lot of time to try understanding what the authors did, and I am still not sure I got everything. Obviously this should not be the case. My general feeling is, anyway, that some decisions were not 100% clear and/or trustable, at least how they were explained now. I might surely have lost something, but I am afraid that the methods are not clear enough and they need to be justified and clarified. For example, authors say that the have 14 species and 10-20 plots for each. But then they say they sampled 108 vegetative plots. Is this incongruence referring to the issue that plots were (moreover) not fully independent spatially? BTW, this is an issue not well taken into account. Hence, the counting (108) is unclear. “Vegetative” is also unclear. Then they say “We removed all individuals of non-focal species” but later they say they have control plots. Introduce better these control plots, please, they come to readers’ surprise. How many plots do they have for each species? Add this into Table 1. A rarefaction for phylogenetic distance indices was done (randomly selecting 11 species in each “community” 1000 times; btw here I do not

know also if community refers to the ~10 plots used for each alien species). I do not understand why simple null-models were not used, just shuffling the identity of the all species removed, out of all plots sampled for a species. And how was done this rarefaction at the regional level? Using also species from different habitats? Also I am not sure why authors need to create 14 community-scale phylogenies, the function 'mpd' and 'mntd' can use a bigger distance matrix than the single plots considered. I am thus a bit scared by the sentence "In this case, branch lengths were rescaled", because different scaling in the 14 species would actually impede any type of comparison between species. What was this rescaling? Why it was needed? What is this modified mpd function doing? All this is unclear, and a bit scary. L302: how was the distance matrix on the circular trait considered (the method by Pavoine et al. 2009 was applied?)? What is the modified version of the Gower distance (L314) doing?

5. I understand that demographic parameters were established only during 1 growing season. Is it so? Is this not this a big problem for the estimation of the demographic models, which should take into account multiple growing seasons? Is it safe to use different types of population model (from Table 1)?

6. The statistical model described starting at line 283, is not very clear, please justify all decisions. I would like to see both results across species (like in the present Fig. 1) but also within species. So, it is not clear if the models in Fig. 1 and Fig. 2 are based on number of observation equal to 14 or to 108. I guess the second is true but, if so, then this means that the authors could present also results of test within species (with some sort of caterpillar plots or so). I understand there is a sort of gradient within the sampling of each species. Anyway a couple of decisions in the statistical test were a bit unclear/obscure. The 14 species are clearly a random selection in the region, hence the absence of species identity (of the alien species) as random factor seems arguable. Also it is not clear why the same model was not also considered for different combination of traits (or single traits), so that Fig. 1 and Table 1 can include also traits as well. Please clarify and show analyses in the main paper.

I finally invite the authors to consider a couple of recent papers:

By Jon Bennett: <https://onlinelibrary.wiley.com/doi/abs/10.1111/jvs.12779>

By Luisa Conti et al.: <https://besjournals.onlinelibrary.wiley.com/doi/abs/10.1111/1365-2745>.

Author's Response to Decision Letter for (RSPB-2019-1695.R0)

See Appendix A.

RSPB-2019-2599.R0

Review form: Reviewer 3

Recommendation

Major revision is needed (please make suggestions in comments)

Scientific importance: Is the manuscript an original and important contribution to its field?

Good

General interest: Is the paper of sufficient general interest?

Acceptable

Quality of the paper: Is the overall quality of the paper suitable?

Good

Is the length of the paper justified?

Yes

Should the paper be seen by a specialist statistical reviewer?

No

Do you have any concerns about statistical analyses in this paper? If so, please specify them explicitly in your report.

Yes

It is a condition of publication that authors make their supporting data, code and materials available - either as supplementary material or hosted in an external repository. Please rate, if applicable, the supporting data on the following criteria.

Is it accessible?

Yes

Is it clear?

Yes

Is it adequate?

Yes

Do you have any ethical concerns with this paper?

No

Comments to the Author

Levin et al tested whether responses to competition of 14 alien plant species were affected by their phylogenetic/ functional traits relatedness to the resident communities at both local and regional scales. It is novel to test the Darwin's naturalization conundrum at different spatial scales. Overall, the writing is clear, and the analyses and interpretation of local scale look good to me. However, the analyses of regional scale did not convince me. Let me explain below.

As briefly introduced by the authors, Darwin's naturalization conundrum might be explained by spatial scales: the naturalization hypothesis is mainly supported in local scales, where competition or other local processes are more important; pre-adaptation is mainly supported at regional scales, where environmental filters are more important (this is well reviewed in Thuiller et al 2010 Diversity and distributions). Therefore, it's not surprising that responses to competition, which is mainly a local process, were not explained by regional-scale novelty. In addition, and more importantly, responses to competition can hardly reveal environmental filters. A more appropriate method is to test the relationship between growth rates in the absence of competition or interspecific competition (i.e. $\lambda_{i,CR}$ in the present paper) and novelty. See P595 in Kraft et al. Functional Ecology 2015, 29, 592-599 for more details.

The 14 alien species are from different habitats. However, the authors calculated the regional-scale novelty with all plant species present in Tyson. By doing so, they included many species that never co-occur with and/or are selected by different environmental filters with the target aliens. Consequently, this method might obscure the 'true' regional-scale novelty and prevent us from testing the environmental filters. If possible to get habitat information for most species, I

suggest to calculate regional-scale with all species that came from the same habitats as the target alien. If not, discuss the limitation.

Minor comments:

The references were not ordered according to their appearance in the text.

L63 'competitive effects' is different with competitive responses (Goldberg, Journal of Ecology 1991, 79, 1013-1030)

L157 It might be confusing to assign those less impactful alien species as naturalized species, because invasive species are also naturalized. I suggest to use non-invasive alien species for them, and mention that all aliens are naturalized.

L180 typo 'n naturalized'

L184 It is not clear to me which species had more than 10 plots. More information will be appreciated. Maybe put it into Table 1 or S1.1.

L131 why some species were parametrized with matrix projection model and others integral projection model?

L234 Why 0.5 was added to the lambdas?

L239 The effect size of competition was set to 0 if the difference between lambdas was not significant. I understand the reason. But this is arbitrary, and could violate the assumptions of linear model, which is hard to test with only 14 data points. I would suggest to test with the original effect sizes. What might help is to add the variance of lambdas into the model, with packages of meta-analyses or gls function in nlme, to give less weight to those lambdas with very high variances.

L253 interactions between aliens PLAY a role...

L474-476 The discussion here does not make sense. Which results revealed the role of niche and fitness differences? In the present version, the authors only calculated the responses of aliens to competition, which is one of the four parameters of niche difference, and one of the six parameters of fitness difference. Please reorganize or delete it.

Table 1. The 'MEPP status' column is not aligned with others. Besides, there are only four non-invasive species in the table, whereas the figures say six.

Review form: Reviewer 4

Recommendation

Accept with minor revision (please list in comments)

Scientific importance: Is the manuscript an original and important contribution to its field?

Excellent

General interest: Is the paper of sufficient general interest?

Good

Quality of the paper: Is the overall quality of the paper suitable?

Good

Is the length of the paper justified?

No

Should the paper be seen by a specialist statistical reviewer?

Yes

Do you have any concerns about statistical analyses in this paper? If so, please specify them explicitly in your report.

No

It is a condition of publication that authors make their supporting data, code and materials available - either as supplementary material or hosted in an external repository. Please rate, if applicable, the supporting data on the following criteria.

Is it accessible?

Yes

Is it clear?

Yes

Is it adequate?

Yes

Do you have any ethical concerns with this paper?

No

Comments to the Author

I read the manuscript by Levin et al. firstly now in the second round of revision. The study deals with an interesting topic. The authors used a set of experiments to disentangle the effect of competition between alien species and native communities. I first impression was that the text is tough to follow. After several readings of the text, I can understand the study design and the results. However, I would strongly recommend to revise again the text.

Especially the methodological part is enormously long. It contains several detail information; some of them are not used further in the results or discussion. They only complicate the text flow. See some examples below.

I would try to unify some terminology. Novel species is for me, newly introduced, and here in the text, it is distantly phylogenetically/functionally related. Alien is here focal species. Competitor is native species (it is strange to accept this point of view).

Aliens are firstly divided into naturalized and invasive species; further, in the text, this point is not mentioned, and authors do not use the status of alien species for any explanation.

The authors work in a large group of habitats from forests to prairies. I wonder if there are no differences in competition rate of single species across habitats.

l. 88-89 I do not agree. Classification of alien status, e.g., causal/naturalized/invasive is not based on the relationship to the native species, but it is based on the distributional characteristic of the species

l. 163-167 detail information, which is useless in the context of the study

l. 171-179 text describes focus on families Fabaceae and Brassicaceae in detail, but looking at table 1, the list of alien species is much broader. Further, the information about the families is not also used in the story. This paragraph could be deleted.

l. 198-201. It is not clear to me if the clipped biomass was somehow used further in the analyses. Please, either give here or to remove this part.

l. 245 phylogeny was pruned to include dicots only. I wonder if this is not a problem in plots from prairies and other non-forest vegetation types, where grasses built an essential part of the native communities. The phylogenetic distance is based on dicots only, and the value could be shifted base on occurrences of monocots.

l. 269-227 it is nice that the authors used several phylotrees to be sure that the results are not affected by the phylogeny. However, such information could be given in the supplementary.

l. 282-284 some of the named traits are further in the text mentioned as too correlated and not used for analyses. Can it be possible to add here only relevant traits?

l. 282 wet/dry ratio - do you mean LDMC?

l. 291 I wonder if the traits were measured for dicots only and if not - Am I right that different species were used for phylogenetic part of analyses and different for functional part? Were the same species used for both the parts of the study?

l. 371-372 probably should be given in the methods.

Review form: Reviewer 5

Recommendation

Major revision is needed (please make suggestions in comments)

Scientific importance: Is the manuscript an original and important contribution to its field?

Acceptable

General interest: Is the paper of sufficient general interest?

Acceptable

Quality of the paper: Is the overall quality of the paper suitable?

Marginal

Is the length of the paper justified?

No

Should the paper be seen by a specialist statistical reviewer?

No

Do you have any concerns about statistical analyses in this paper? If so, please specify them explicitly in your report.

Yes

It is a condition of publication that authors make their supporting data, code and materials available - either as supplementary material or hosted in an external repository. Please rate, if applicable, the supporting data on the following criteria.

Is it accessible?

Yes

Is it clear?

Yes

Is it adequate?

Yes

Do you have any ethical concerns with this paper?

No

Comments to the Author

In the paper entitled 'Phylogenetic and functional novelty explain alien plant population responses to competition' the authors aim to find support for darwin's naturalization hypothesis, i.e. that evolutionary distinct alien plants should compete less with the resident flora. To do this, the authors use a removal experiment as well as parametrize matrix population and integral projection models, along with phylogenetic and trait information. They found that alien species that were more distinct to their competitors were indeed less affected by competition, although this pattern was not maintained when calculating relatedness at the regional level. They also found that the models explaining competition effects were improved when including functional distinctiveness. The authors argue that their findings support the naturalization hypothesis and they highlight both the scale and the functional vs phylogenetic information issues.

Generally, this paper was very hard to follow and needs much improvement in terms of structure and delivery of the message. I do think that the results are interesting and experimental evidence

is surely needed, but in its present form the paper is not suited for publication. I have three major concerns which I explain below.

First, the authors fail to clearly state their aims and hypothesis (e.g. paragraphs at the end of the introduction in LL 134-148 describe methods and results). As a consequence, the whole text lacks a structure, with concepts presented in a random manner, and it would need major editing. The authors seemed to have slightly ameliorated the manuscript from the first revision. However, even though I did not perform the first revision, my opinion is that these changes were mostly superficial. What are exactly the questions that the authors are asking? What are the expectations regarding those questions?

Second, it is unclear what the authors mean by 'alien success'. This derives from the fact that they do not explain early in the text the exact invasive stage at which they are working. They do mention several times that they work on the post-establishment phase, but I) they seem to want to test the Darwin's naturalization hypothesis, which as the name suggests, deals with naturalization, II) their definition of naturalized 'currently considered relatively benign' potentially includes also casual species (that are not able to maintain viable populations and therefore are not considered established), III) they discuss propagule pressure and environmental filtering, which are mechanisms known to influence the establishment phase rather than the spreading phase. The mechanism they directly assess, competition, is present at both stages. What invasion stage are the authors studying? What are their expectations regarding the role/importance of competition in that particular stage?

Finally, I found the methods rather complex and not explained sufficiently. It is rather difficult for the reader to understand what was done and why. For example, it was not clear to me what was done exactly with the trait information and to what purpose, nor why the comparison between the functional and phylogenetic information was needed. The models seem far too complex (or maybe not explained clearly). Why not use a single full model with all functional and phylogenetic metrics, select the best model, and discuss the variables maintained?

Decision letter (RSPB-2019-2599.R0)

10-Dec-2019

I am writing to inform you that this version of your manuscript RSPB-2019-2599 entitled "Phylogenetic and functional novelty explain alien plant population responses to competition" has, in its current form, been rejected for publication in Proceedings B.

This action has been taken on the advice of referees, who have recommended that substantial revisions are necessary. With this in mind we would be happy to consider a resubmission, provided the comments of the referees are fully addressed. However please note that this is not a provisional acceptance. Please also note that I have made an exception in allowing a second round of major revision. In by far the most cases, manuscripts that fail to sufficiently converge after a round of major revision are rejected for Proceedings B. Given the potential interest in your results, and in light of the current enthusiasm of the Associate Editor, I agree that a second chance at convincing the reviewers is called for. There should be sufficient convergence though because otherwise the manuscript may still be rejected in the next round.

Please find below the comments made by the referees, not including confidential reports to the Editor, which I hope you will find useful.

Sincerely,
 Professor Hans Heesterbeek
 mailto: proceedingsb@royalsociety.org

Associate Editor Board Member

Comments to Author:

We now have three thoughtful reviews of this round of the manuscript. The reviews which are quite extensive have one main message: the current manuscript is "very hard to follow and needs much improvement in terms of structure and delivery of the message" in the words of reviewer 3. I, like the reviewers, think this experiment is very interesting and deserves a large audience in the field, but the similarities of the reviewers reactions implies that there is still a lot more work to do on clarifying the manuscript. I won't add more to the extensive reviewer comments, but I do think they represent the reactions of the ideal audience for this work, and as such the authors should take their comments very seriously in a revision.

Reviewer(s)' Comments to Author:

Referee: 3

Comments to the Author(s).

Levin et al tested whether responses to competition of 14 alien plant species were affected by their phylogenetic/ functional traits relatedness to the resident communities at both local and regional scales. It is novel to test the Darwin's naturalization conundrum at different spatial scales. Overall, the writing is clear, and the analyses and interpretation of local scale look good to me. However, the analyses of regional scale did not convince me. Let me explain below.

As briefly introduced by the authors, Darwin's naturalization conundrum might be explained by spatial scales: the naturalization hypothesis is mainly supported in local scales, where competition or other local processes are more important; pre-adaptation is mainly supported at regional scales, where environmental filters are more important (this is well reviewed in Thuiller et al 2010 Diversity and distributions). Therefore, it's not surprising that responses to competition, which is mainly a local process, were not explained by regional-scale novelty. In addition, and more importantly, responses to competition can hardly reveal environmental filters. A more appropriate method is to test the relationship between growth rates in the absence of competition or interspecific competition (i.e. $\lambda_{i,CR}$ in the present paper) and novelty. See P595 in Kraft et al. Functional Ecology 2015, 29, 592-599 for more details.

The 14 alien species are from different habitats. However, the authors calculated the regional-

scale novelty with all plant species present in Tyson. By doing so, they included many species that never co-occur with and/or are selected by different environmental filters with the target aliens. Consequently, this method might obscure the 'true' regional-scale novelty and prevent us from testing the environmental filters. If possible to get habitat information for most species, I suggest to calculate regional-scale with all species that came from the same habitats as the target alien. If not, discuss the limitation.

Minor comments:

The references were not ordered according to their appearance in the text.

L63 'competitive effects' is different with competitive responses (Goldberg, *Journal of Ecology* 1991, 79, 1013-1030)

L157 It might be confusing to assign those less impactful alien species as naturalized species, because invasive species are also naturalized. I suggest to use non-invasive alien species for them, and mention that all aliens are naturalized.

L180 typo 'n naturalized'

L184 It is not clear to me which species had more than 10 plots. More information will be appreciated. Maybe put it into Table 1 or S1.1.

L131 why some species were parametrized with matrix projection model and others integral projection model?

L234 Why 0.5 was added to the lambdas?

L239 The effect size of competition was set to 0 if the difference between lambdas was not significant. I understand the reason. But this is arbitrary, and could violate the assumptions of linear model, which is hard to test with only 14 data points. I would suggest to test with the original effect sizes. What might help is to add the variance of lambdas into the model, with packages of meta-analyses or `gls` function in `nlme`, to give less weight to those lambdas with very high variances.

L253 interactions between aliens PLAY a role...

L474-476 The discussion here does not make sense. Which results revealed the role of niche and fitness differences? In the present version, the authors only calculated the responses of aliens to competition, which is one of the four parameters of niche difference, and one of the six parameters of fitness difference. Please reorganize or delete it.

Table 1. The 'MEPP status' column is not aligned with others. Besides, there are only four non-invasive species in the table, whereas the figures say six.

Referee: 4

Comments to the Author(s).

I read the manuscript by Levin et al. firstly now in the second round of revision. The study deals with an interesting topic. The authors used a set of experiments to disentangle the effect of competition between alien species and native communities. My first impression was that the text is tough to follow. After several readings of the text, I can understand the study design and the results. However, I would strongly recommend to revise again the text.

Especially the methodological part is enormously long. It contains several detail information; some of them are not used further in the results or discussion. They only complicate the text flow. See some examples below.

I would try to unify some terminology. Novel species is for me, newly introduced, and here in the text, it is distantly phylogenetically/functionally related. Alien is here focal species. Competitor is native species (it is strange to accept this point of view).

Aliens are firstly divided into naturalized and invasive species; further, in the text, this point is not mentioned, and authors do not use the status of alien species for any explanation.

The authors work in a large group of habitats from forests to prairies. I wonder if there are no differences in competition rate of single species across habitats.

L. 88-89 I do not agree. Classification of alien status, e.g., causal/naturalized/invasive is not based on the relationship to the native species, but it is based on the distributional characteristic of the species

- l. 163-167 detail information, which is useless in the context of the study
- l. 171-179 text describes focus on families Fabaceae and Brassicaceae in detail, but looking at table 1, the list of alien species is much broader. Further, the information about the families is not also used in the story. This paragraph could be deleted.
- l. 198-201. It is not clear to me if the clipped biomass was somehow used further in the analyses. Please, either give here or to remove this part.
- l. 245 phylogeny was pruned to include dicots only. I wonder if this is not a problem in plots from prairies and other non-forest vegetation types, where grasses built an essential part of the native communities. The phylogenetic distance is based on dicots only, and the value could be shifted base on occurrences of monocots.
- l. 269-227 it is nice that the authors used several phylotrees to be sure that the results are not affected by the phylogeny. However, such information could be given in the supplementary.
- l. 282-284 some of the named traits are further in the text mentioned as too correlated and not used for analyses. Can it be possible to add here only relevant traits?
- l. 282 wet/dry ratio - do you mean LDMC?
- l. 291 I wonder if the traits were measured for dicots only and if not - Am I right that different species were used for phylogenetic part of analyses and different for functional part? Were the same species used for both the parts of the study?
- l. 371-372 probably should be given in the methods.

Referee: 5

Comments to the Author(s).

In the paper entitled 'Phylogenetic and functional novelty explain alien plant population responses to competition' the authors aim to find support for darwin's naturalization hypothesis, i.e. that evolutionary distinct alien plants should compete less with the resident flora. To do this, the authors use a removal experiment as well as parametrize matrix population and integral projection models, along with phylogenetic and trait information. They found that alien species that were more distinct to their competitors were indeed less affected by competition, although this pattern was not maintained when calculating relatedness at the regional level. They also found that the models explaining competition effects were improved when including functional distinctiveness. The authors argue that their findings support the naturalization hypothesis and they highlight both the scale and the functional vs phylogenetic information issues.

Generally, this paper was very hard to follow and needs much improvement in terms of structure and delivery of the message. I do think that the results are interesting and experimental evidence is surely needed, but in its present form the paper is not suited for publication. I have three major concerns which I explain bellow.

First, the authors fail to clearly state their aims and hypothesis (e.g. paragraphs at the end of the introduction in LL 134-148 describe methods and results). As a consequence, the whole text lacks a structure, with concepts presented in a random manner, and it would need major editing. The authors seemed to have slightly ameliorated the manuscript from the first revision. However, even though I did not perform the first revision, my opinion is that these changes were mostly superficial. What are exactly the questions that the authors are asking? What are the expectations regarding those questions?

Second, it is unclear what the authors mean by 'alien success'. This derives from the fact that they do not explain early in the text the exact invasive stage at which they are working. They do mention several times that they work on the post-establishment phase, but I) they seem to want to test the Darwin's naturalization hypothesis, which as the name suggests, deals with naturalization, II) their definition of naturalized 'currently considered relatively benign' potentially includes also casual species (that are not able to maintain viable populations and therefore are not considered established), III) they discuss propagule pressure and environmental filtering, which are mechanisms known to influence the establishment phase rather than the

spreading phase. The mechanism they directly assess, competition, is present at both stages. What invasion stage are the authors studying? What are they expectations regarding the role/importance of competition in that particular stage?

Finally, I found the methods rather complex and not explained sufficiently. It is rather difficult for the reader to understand what was done and why. For example, it was not clear to me what was done exactly with the trait information and to what purpose, nor why the comparison between the functional and phylogenetic information was needed. The models seem far too complex (or maybe not explained clearly). Why not use a single full model with all functional and phylogenetic metrics, select the best model, and discuss the variables maintained?

Author's Response to Decision Letter for (RSPB-2019-2599.R0)

See Appendix B.

RSPB-2020-1070.R0

Review form: Reviewer 3

Recommendation

Accept with minor revision (please list in comments)

Scientific importance: Is the manuscript an original and important contribution to its field?

Good

General interest: Is the paper of sufficient general interest?

Good

Quality of the paper: Is the overall quality of the paper suitable?

Good

Is the length of the paper justified?

Yes

Should the paper be seen by a specialist statistical reviewer?

No

Do you have any concerns about statistical analyses in this paper? If so, please specify them explicitly in your report.

Yes

It is a condition of publication that authors make their supporting data, code and materials available - either as supplementary material or hosted in an external repository. Please rate, if applicable, the supporting data on the following criteria.

Is it accessible?

Yes

Is it clear?

Yes

Is it adequate?

Yes

Do you have any ethical concerns with this paper?

No

Comments to the Author

I thank the authors for addressing my comments. I have only a few specific comments now. I encourage the authors to discuss why the relationship between distinctiveness and the effect size of competition disappeared at large scale. Blanchet et al 2020 Ecology Letters (10.1111/ele.13525) might help.

L56 use 'because' rather than 'due to' to keep the form parallel.

L71 & L406-407 Here, the authors want to test why some aliens become invasive. But this requires to compare non-invasive aliens with invasive aliens. I would replace 'invasive' with 'successful' (or something similar) unless the authors include non-invasive vs invasive in the models. Besides, the authors have removed the categories of alien species (i.e. invasive vs non-invasive) in the current version. I would still keep them, otherwise some audience may think all species are invasive.

L86 replace 'herbivore release' with 'herbivores'. If aliens are enemy released, there will be no indirect effects mediated by enemies.

L87 To the best of my knowledge, Elton hadn't discussed (or even mentioned) Darwin's naturalization hypothesis in his 1958 book. I apologize if I am wrong.

L346 & 348 It will help to write the abbreviation in full form at the beginning of each section, as well as in the figure legends.

L418 What does 'invasion maturity' mean? Consider to reword.

Decision letter (RSPB-2020-1070.R0)

01-Jun-2020

Dear Mr Levin

I am pleased to inform you that your manuscript RSPB-2020-1070 entitled "Phylogenetic and functional distinctiveness explain alien plant population responses to competition" has been accepted for publication in Proceedings B.

The referee has recommended publication, but also suggests some minor revisions to your manuscript. Therefore, I invite you to respond to the referee's comments and revise your manuscript. Because the schedule for publication is very tight, it is a condition of publication that you submit the revised version of your manuscript within 7 days. If you do not think you will be able to meet this date please let us know.

When submitting your revised manuscript, you will be able to respond to the comments made by the referee(s) and upload a file "Response to Referees". You can use this to document any changes you make to the original manuscript. We require a copy of the manuscript with revisions made

since the previous version marked as 'tracked changes' to be included in the 'response to referees' document.

Sincerely,
 Professor Hans Heesterbeek
 mailto: proceedingsb@royalsociety.org

Associate Editor

Comments to Author:

Thank you for your careful attention to the previous round of reviews. The reviewer has requested some additional small changes, which should further improve the manuscript. This manuscript has the potential to make a real contribution to the literature in this area, but there are a number of details that need attention before that can happen.

Reviewer(s)' Comments to Author:

Referee: 3

Comments to the Author(s).

I thank the authors for addressing my comments. I have only a few specific comments now. I encourage the authors to discuss why the relationship between distinctiveness and the effect size of competition disappeared at large scale. Blanchet et al 2020 Ecology Letters (10.1111/ele.13525) might help.

L56 use 'because' rather than 'due to' to keep the form parallel.

L71 & L406-407 Here, the authors want to test why some aliens become invasive. But this requires to compare non-invasive aliens with invasive aliens. I would replace 'invasive' with 'successful' (or something similar) unless the authors include non-invasive vs invasive in the models. Besides, the authors have removed the categories of alien species (i.e. invasive vs non-invasive) in the current version. I would still keep them, otherwise some audience may think all species are invasive.

L86 replace 'herbivore release' with 'herbivores'. If aliens are enemy released, there will be no indirect effects mediated by enemies.

L87 To the best of my knowledge, Elton hadn't discussed (or even mentioned) Darwin's naturalization hypothesis in his 1958 book. I apologize if I am wrong.

L346 & 348 It will help to write the abbreviation in full form at the beginning of each section, as well as in the figure legends.

L418 What does 'invasion maturity' mean? Consider to reword.

Author's Response to Decision Letter for (RSPB-2020-1070.R0)

See Appendix C.

Decision letter (RSPB-2020-1070.R1)

05-Jun-2020

Dear Mr Levin

I am pleased to inform you that your manuscript entitled "Phylogenetic and functional distinctiveness explain alien plant population responses to competition" has been accepted for publication in Proceedings B.

Your article has been estimated as being 9 pages long. Our Production Office will be able to confirm the exact length at proof stage.

Open Access

Paper charges

Sincerely,

Proceedings B

Appendix A

Dear Dr. Hesterbeek,

Thank you for the very helpful review of our manuscript RSPB-2019-1695 entitled "Phylogenetic and functional novelty explain alien plant population responses to competition". We appreciate the thoughtful comments by the Associate Editor and reviewers. We feel that these critiques have vastly strengthened this manuscript. In the revised version, we have made appropriate revisions throughout the manuscript and addressed the comments in full. Please see below a point-by-point description of how we have addressed each comment (*in italics*).

Sincerely,

Sam Levin, Raelene Crandall, Tyler Pokoski, Claudia Stein & Tiffany Knight

Specific responses

Associate Editor

Board Member: 1

Comments to Author:

We now have two reviews of this paper on alien plant population responses to competition. Both reviewer--and i agree--found this to be an interesting experimental addition to a mostly observational literature. There is certainly something here to offer. However, both reviewers also found the introduction to be clear but the methods and results to be very confusing. I agree. I liked the shinyapp, but the fact the complexity of the results needs this level of interactivity is and will be confusing for readers. Furthermore, the connection back to theory gets very hazy. The reviewers both have suggestions for clarifying specific parts, but in my view this requires a more substantial re-thinking about which are the important results here, and then streamlining the paper to clearly build to those results.

We have worked to make our paper less complex to read. We now present additional information in our methods and results, that should better link the conceptual framework, methods and results (a new conceptual figure, Figure 1, lines 157-159, 221, 246-247, and others listed below in reference to specific comments from the reviewers). We have also added a conceptual figure (the new Figure 1). We admit that the analyses in this paper are complex and multi-staged, but we would also argue that these are necessary. Much of the controversy surrounding this topic has resulted from papers having a too simplistic approach – likely because authors are making the most of the currently available data. We do not make this point in the introduction because we feel we explain why the analyses are needed in our methods.

We have restructured the discussion to include more on the relationship between our study and invasion stage and coexistence (L 437-490).

One detailed comment: square-root of phylogenetic distance rather than the raw values are a much better match for reasonable models of functional difference between species for reasons laid out in this paper:

Letten, Andrew D., and William K. Cornwell. "Trees, branches and (square) roots: why evolutionary relatedness is not linearly related to functional distance." *Methods in Ecology and Evolution* 6.4 (2015): 439-444.

This is an excellent point! We have re-run all of our analyses using square root transformed branch lengths (L 246-247). Note that the results do not change substantively (L 376-383).

Reviewer(s)' Comments to Author:

Referee: 1

Comments to the Author(s)

There have been various test of Darwin's Naturalisation Hypothesis but few have taken an experimental approach as in this study and thus this work does merit attention. Nevertheless, the authors made the the manuscript quite hard going. It was hard to navigate the manuscript without details of the models and calculation of lambda. I checked out the online supplement but feel that this still made it hard work to see exactly how the models were put together and lambda calculated. My assumption is that lambda was calculated as the number of individuals in the plot at time t+1 compared to time t but some of the plot sizes, especially for woody species, would be too small to calculate such a measure. The authors therefore need to provide a section on the demographic modelling approach and how the models were parameterized and how aspects like seedbank and fecundity were estimated by the authors. Without this detail it is hard to see exactly what was done and the robustness of their conclusions. Obviously this lengthens the manuscript and if material needs to be cut then perhaps it should related to the scaling issue, which is well known by now and perhaps is the less informative aspect in this manuscript.

Lambda was computed as the dominant eigenvalue of the projection matrices (Caswell 2001, Easterling et al. 2000, Ellner, Rees & Childs 2016). This is standard methodology, and thus it does not take much additional space to describe the approach. We have added this information to the Methods section (L 221) and apologize for that oversight.

Minor comments:

I would suggest the authors use the standard terms alien, invasive and naturalized and not confuse matters by using exotic in a way that has heretofore not been used.

We agree with the suggestion to harmonize terminology across studies. Species are now classified as alien (all), invasive (damaging), and naturalized (benign) (L 157-159).

Referee: 2

Comments to the Author(s)

This is a potentially interesting study assessing the effect of removal of competition on the spread of alien species within a region. The authors have sampled vegetation in different plots (~10-20), across different habitats, for each of the 14 alien species selected. They removed all competitors in this plots and assessed changes in demographic parameters as a measure of the effect of competition on alien species (but, to my understanding, this was done only over one

growing season). The effect of competition was significant on ~50% of the 14 species considered, generally only on invasive species, i.e. those which historically have been expanding more in the region (the authors distinguish invasive vs. exotic, where exotic have expanded less and only to disturbed conditions; i.e. invasive + exotic = alien, according to the terminology used in this study). The effect of competition was also stronger when the aliens were more phylogenetically similar species to the set of species removed (notice that here I use “species removed” and I haven’t said “native”, because the species removed were not always native; this aspect was not very clearly taken into account in the study). Measures of functional dissimilarity, based on some on-site trait measurements + bibliographic information, were also considered. However, functional dissimilarity alone was, if I understand well the results, not a good predictor when used alone (although I would like to see Fig. 1 and Table 2, also for different set of traits). Functional traits, at the same time, seem to improve some predictions based only on phylogenetic distance between species (based on the method proposed by Cadotte et al. 2013). Two measures of phylogenetic and functional diversity were considered, MPD and MNTD, and only on the second the combination of functional and phylogenetic information was relevant.

I have some major problems with study, in its present form:

1. Of course, as the authors already partially recognize, the study focus “only” on the effect of competition on the spreading of alien species, when they have already established. The results should be better put into a proper context. Also, removal experiment generally focus on the removal of 1 or 2 dominant species, while here the authors remove all species, which is more a simulation of disturbance, to me, than competition alone. See also next point.

We have added additional information to clarify that this is on the spread/transition to dominance stage of invasions, rather than establishment (L 437 – 470).

2. Following point 1, my hypothesis (before looking at the results) was that “competition” effects would be stronger on exotic species, i.e. those that are assumed to spread less because they are found only on disturbed sites (LL166-170). Hence the results could be seen also as a bit counterintuitive, i.e. the effect of competition was actually stronger on species that actually were already able to win the competition with native species, i.e. they have already expanded in the region. Clear hypotheses on the effect of competition on “invasive” vs. “exotic” species should be better introduced and discussed, particularly with the type of experiment considered.

We agree that our original terminology was not clear – the species with significant responses to competition were the exotic/naturalized ones, not the invasive ones. This has been a common critique, and so we have adopted an alternative terminology: alien, invasive, and naturalized (which replaces exotic) and updated all occurrences in the paper, SI, and figure captions. We have also added a conceptual diagram that explains how the results should be interpreted. This is now Figure 1, and the other figures have become Figure 2 and 3.

3. Clearly the design is not well balanced in terms of phylogenetic distances considered.

When compared to species removed, the values of phylogenetic distance of invasive species (Fig. 1) is likely smaller and more variable than the ones of exotic species. Is there a similar effect on trait dissimilarity? please show us. This seems to be a serious issue, for both MPD and MNTD. This problem cannot be solved now, but anyway the authors say in the methods that they tried to balance the selection of some invasive and exotic species within families. Where are these comparisons? Not in the results section, at least.

We did our best to sample invasive and exotic species across a range of phylogenetic distances. This study considers 14 species from 10 families. In the Brassicaceae, we were able to sample both naturalized and invasive species, which we purposely did for balance. Unfortunately, with only $N = 3$ Brassicaceae species and $N = 2$ Fabaceae species, statistical comparisons within a family would be unreliable. We have added some additional text to the revised manuscript to clarify the underlying idea behind this choice of paired sampling within families in the methods section (L173-175).

These results are potentially important because they can validate, or not, the general results from this study. BTW, most of the results seem to be affected by one or two alien species, one being a tree, which was quite unique phylogenetically (and functionally). How much this/se species influence the results? Should we trust the generality of the results?

As mentioned above, we have revised our figure legends to clarify the invasive/naturalized terminology. Furthermore, we address the question of explanatory power of traits alone in Figure 3 (e.g. R^2_{adj} values when $a = 0$).

To examine the potential for our results to be driven by the presence of two alien species that are rather phylogenetically (and functionally) unique/similar, we have re-run the analyses without the outliers mentioned. The results do not change (Please see Figure S2.8 in Appendix 2).

4. The description of methods is certainly not very clear, and need clear improvements. I had to spend a lot of time to try understanding what the authors did, and I am still not sure I got everything. Obviously this should not be the case. My general feeling is, anyway, that some decisions were not 100% clear and/or trustable, at least how they were explained now. I might surely have lost something, but I am afraid that the methods are not clear enough and they need to be justified and clarified. For example, authors say that they have 14 species and 10-20 plots for each. But then they say they sampled 108 vegetative plots. Is this incongruence referring to the issue that plots were (moreover) not fully independent spatially? BTW, this is an issue not well taken into account. Hence, the counting (108) is unclear. "Vegetative" is also unclear. Then they say "We removed all individuals of non-focal species" but later they say they have control plots. Introduce better these control plots, please, they come to readers' surprise. How many plots do they have for each species? Add this into Table 1.

We have added text to better describe our methods. We specifically clarify how the treatments were applied (L 189-201). Further, we have added a conceptual figure to illustrate the methods graphically. This is now Figure 1, and the other figures have become Figure 2 and 3.

A rarefaction for phylogenetic distance indices was done (randomly selecting 11 species in each “community” 1000 times; btw here I do not know also if community refers to the ~10 plots used for each alien species). I do not understand why simple null-models were not used, just shuffling the identity of the all species removed, out of all plots sampled for a species. And how was done this rarefaction at the regional level? Using also species from different habitats? Also I am not sure why authors need to create 14 community-scale phylogenies, the function ‘mpd’ and ‘mntd’ can use a bigger distance matrix than the single plots considered.

We did not use the functions mpd and mntd because our data were structured as a species list with abundances and sites recorded in single columns rather than the standard site x species matrix (i.e. in “long” format rather than “wide” format), which can be inefficient for graphing and modeling (e.g. with ggplot2, glm). Instead, we used slightly modified forms that accepted our data structure as arguments but are mathematically equivalent. Our source code is open source and available at the repository cited in the text.

I am thus a bit scared by the sentence “In this case, branch lengths were rescaled”, because different scaling in the 14 species would actually impede any type of comparison between species. What was this rescaling? Why it was needed? What is this modified mpd function doing? All this is unclear, and a bit scary.

It was a poor choice of words to use “rescaled”. We have updated that to reflect that branch lengths were weighted by abundances, which is standard practice (L 263-266).

L302: how was the distance matrix on the circular trait considered (the method by Pavoine et al. 2009 was applied)? What is the modified version of the Gower distance (L314) doing?

The modified version of the Gower distance is described in the Pavoine et al. (2009) paper, which this reviewer mentioned. We cite this paper in the text. We are unsure how to make this clearer, but are amenable to changes if the editor deems it necessary.

5. I understand that demographic parameters were established only during 1 growing season. Is it so? Is this not this a big problem for the estimation of the demographic models, which should take into account multiple growing seasons? Is it safe to use different types of population model (from Table 1)?

Demographic parameters were estimated from two growing seasons. This has been clarified (L 211-212).

All species were modeled using identical model forms for each treatment, so computation of effect sizes is not problematic in any way. Models were chosen to include the appropriate number of continuous and discrete state variables that describe the life history of each species. There is a supporting literature in demography on using this approach to ask general questions across species,

including meta-analyses that use global data across the vascular plant tree of life (e.g. Salguero-Gomez et al. 2012). We cite this literature in our paper as justification for our comparisons (L 223-225).

6. The statistical model described starting at line 283, is not very clear, please justify all decisions. I would like to see both results across species (like in the present Fig. 1) but also within species. So, it is not clear if the models in Fig. 1 and Fig. 2 are based on number of observation equal to 14 or to 108. I guess the second is true but, if so, then this means that the authors could present also results of test within species (with some sort of caterpillar plots or so).

The results across species are the results shown in the paper (i.e. number of observations = 14). Within species comparisons are not possible because data were pooled for all species x treatment combinations (i.e., there is only one data point per species). We have added text to clarify this in the methods (L 296-297).

I understand there is a sort of gradient within the sampling of each species.

We are unsure which gradient is referred to here. If the reviewer clarifies their comment we would be happy to address it.

Anyway a couple of decisions in the statistical test were a bit unclear/obscure. The 14 species are clearly a random selection in the region, hence the absence of species identity (of the alien species) as random factor seems arguable.

Including species as a random effect is not possible, as there is only one data point per species (see point above re: number of plots).

Also it is not clear why the same model was not also considered for different combination of traits (or single traits), so that Fig. 1 and Table 1 can include also traits as well. Please clarify and show analyses in the main paper.

We did test single traits and combinations of traits, but they are never the best models in terms of variance explained. For example, first flowering and growth form combined without phylogeny explain ~45% of the variance in effect size of competition. However, the best model for that combination of traits weights phylogeny 50% and explains ~75% of the variance in competition (less than what was reported in the main text). Reporting results for every combination of traits is not really feasible, since there are 7 traits and 127 different unique combinations of them. Due to constraints on the length of the main text, we have added trait-only versions of Figure 2 for selected combinations of traits to Appendix 3.

Appendix B

Dear Professor Hans Heesterbeek,

We appreciate the opportunity for a second round of major revisions, especially since this is rarely allowed in *Proceedings B*. We appreciate the enthusiasm of the Associate Editor and reviewers of our work, and have taken care to incorporate all of the comments into our revision. Please see our revised version (RSPB-2019-2599), as well as our response to the Associate Editor and reviewers' comments in italics below.

Warm regards,
Sam Levin (on behalf of all co-authors)

Associate Editor Board Member

We now have three thoughtful reviews of this round of the manuscript. The reviews which are quite extensive have one main message: the current manuscript is "very hard to follow and needs much improvement in terms of structure and delivery of the message" in the words of reviewer 3. I, like the reviewers, think this experiment is very interesting and deserves a large audience in the field, but the similarities of the reviewers reactions implies that there is still a lot more work to do on clarifying the manuscript. I won't add more to the extensive reviewer comments, but I do think they represent the reactions of the ideal audience for this work, and as such the authors should take their comments very seriously in a revision.

Thank you for the opportunity to revise our manuscript. We made major changes to improve the structure and delivery of the message. After this round of review, what we needed to do became quite clear to us, and we took it very seriously. We have almost completely re-written the introduction, and the organization of the methods also had a major reconstruction. Our previous version of the introduction was framed too much under the umbrella of Darwin's Naturalization Conundrum. This leads the reader to think that we will directly address the role of phylogenetic and functional distinctiveness in the environmental filtering process, which we do not do. Our new introduction focuses on the conceptual framework for what we test rigorously in our study—the role of distinctiveness in the strength of competition with resident plants. We have reorganized the presentation of the methods and streamlined the methods (e.g., we don't test for different responses of invasive and non-invasive alien plants, and so there was no need to have information about how to categorize aliens), and updated our discussion accordingly.

Referee: 3

Levin et al tested whether responses to competition of 14 alien plant species were affected by their phylogenetic/ functional traits relatedness to the resident communities at both local and regional scales. It is novel to test the Darwin's naturalization conundrum at different spatial scales. Overall, the writing is clear, and the analyses and interpretation of local scale look good to me. However, the analyses of regional scale did not convince me. Let me explain below.

As briefly introduced by the authors, Darwin's naturalization conundrum might be explained by spatial scales: the naturalization hypothesis is mainly supported in local scales, where competition or other local processes are more important; pre-adaptation is mainly supported at regional scales, where environmental filters are more important (this is well reviewed in Thuiller et al 2010 Diversity and distributions). Therefore, it's not surprising that responses to competition, which is mainly a local process, were not explained by regional-scale novelty. In addition, and more importantly, responses to competition can hardly reveal environmental filters. A more appropriate method is to test the relationship between growth rates in the absence of competition or

interspecific competition (i.e. $\lambda_{i,CR}$ in the present paper) and novelty. See P595 in Kraft et al. *Functional Ecology* 2015, 29, 592–599 for more details.

The 14 alien species are from different habitats. However, the authors calculated the regional-scale novelty with all plant species present in Tyson. By doing so, they included many species that never co-occur with and/or are selected by different environmental filters with the target aliens. Consequently, this method might obscure the 'true' regional-scale novelty and prevent us from testing the environmental filters. If possible to get habitat information for most species, I suggest to calculate regional-scale with all species that came from the same habitats as the target alien. If not, discuss the limitation.

We framed the introduction of the manuscript under the concepts surrounding Darwin's naturalization conundrum and spatial scale. And, as correctly pointed out by this reviewer, we did not provide an appropriate test of the roles of competition vs environmental filtering. Our experiment is uniquely designed to test the role of alien distinctiveness on the strength of competition with the resident community. Our strength is our measure of performance, which involved lifetime fitness of our focal alien species. We are not designed to examine the environmental filtering processes, which would ideally require data on the habitat associations for all species in our regional spatial grain. We have responded to this comment not by adding a new analysis on environmental filters, but instead by completely re-framing our introduction.

We delete the focus on Darwin's naturalization conundrum, and our introduction now has two new paragraphs:

- 1) We introduce a prominent category of hypotheses invoked to explain invasiveness that involves the strength of negative interspecific interactions with the resident community. The idea here is that resident community should provide weak resistance to alien species that are functionally distinct, allowing these species to have high performance.*
- 2) We introduce that despite its prominence in the literature, empirical support for the notion that more distinct species should have higher performance has been mixed. There are four possible reasons for this: (i) Coexistence and community assembly theory, (ii) Methodological differences across studies in the measures of distinctiveness (phylogenetic and functional are usually considered separately). (iii) multiple types of data are often employed to test hypotheses (presence-absence data, relative abundance data, and performance data) and, (iv) the spatial grain in which the resident community is defined.*

We now justify that we test whether the result found at a small spatial grain is robust even if the resident community is defined at a larger spatial grain that would be typical for studies using species checklists. Many studies use such checklists, and invoke competition as a likely mechanism to explain their results. This might be a fine thing to do, if the result we find is robust to the spatial grain in which the resident community is defined. This justification for the regional scale analysis is much more appropriate than our previous text, which was framed under environmental filtering.

Minor comments:

The references were not ordered according to their appearance in the text.

They are now ordered by appearance, rather than alphabetically as they were before.

L63 'competitive effects' is different with competitive responses (Goldberg, *Journal of Ecology* 1991, 79, 1013-1030)

This was a typo and is now corrected.

L157 It might be confusing to assign those less impactful alien species as naturalized species, because invasive species are also naturalized. I suggest to use non-invasive alien species for them, and mention that all aliens are naturalized.

We have removed the categories of alien species, as we do not provide an explicit test that uses these categories. This was one of the places in which our methods were needlessly complex and caused confusion.

L180 typo 'n naturalized'

This text no longer exists.

L184 It is not clear to me which species had more than 10 plots. More information will be appreciated. Maybe put it into Table 1 or S1.1.

This information is now added to Table 1

L131 why some species were parametrized with matrix projection model and others integral projection model?

We have added justification for this to the main text (L183-187 and L196-198). Some species' demography is best described with discrete stage/age/size classification. MPMs are the appropriate tool for these. Others, primarily trees, are best described using continuous state variables and require an IPM.

L234 Why 0.5 was added to the lambdas?

*Some models generated lambdas that were very close to 0 in the control treatment (i.e. *Lepidium campestre*). This is problematic, as the log of 0 is undefined. We have added a line to explain this in the manuscript and included a reference (L208-209).*

L239 The effect size of competition was set to 0 if the difference between lambdas was not significant. I understand the reason. But this is arbitrary, and could violate the assumptions of linear model, which is hard to test with only 14 data points. I would suggest to test with the original effect sizes. What might help is to add the variance of lambdas into the model, with packages of meta-analyses or gls function in nlme, to give less weight to those lambdas with very high variances.

We appreciate this comment and agree that it is important to quantify how uncertainty in our demography translates into uncertainty in our regressions. We did not do this before. We have re-done the entire phylogenetic portion of the analysis, and describe methods in detail in L270-277.

To summarize, we took the lambda values from the MPM/IPM bootstrapping procedure, and computed 1000 effect sizes of competition for each species. We then re-fit the regression models 1000 times using these 1000 unique values. We present the distributions of the coefficients associated with all 6 distinctiveness metrics, and highlight the fact that at the small grain, novelty coefficients were never 0 or positive, while at the regional grain, 0 is included in 95% CIs. We also present the observed R^2_{adj} values and their 95% CIs. These panels are combined to create a new version of Figure 2. We feel

that this addresses the concern of this reviewer – that we need to propagate uncertainty in demographic data through the rest of our analysis to ensure that we are seeing a real effect and not a sampling artifact.

L253 interactions between aliens PLAY a role...

This typo has been corrected.

L474-476 The discussion here does not make sense. Which results revealed the role of niche and fitness differences? In the present version, the authors only calculated the responses of aliens to competition, which is one of the four parameters of niche difference, and one of the six parameters of fitness difference. Please reorganize or delete it.

We removed paragraph about niche vs. fitness differences.

Table 1. The 'MEPP status' column is not aligned with others. Besides, there are only four non-invasive species in the table, whereas the figures say six.

This column is now removed

Referee: 4

I read the manuscript by Levin et al. firstly now in the second round of revision. The study deals with an interesting topic. The authors used a set of experiments to disentangle the effect of competition between alien species and native communities. I first impression was that the text is tough to follow. After several readings of the text, I can understand the study design and the results. However, I would strongly recommend to revise again the text. Especially the methodological part is enormously long. It contains several detail information; some of them are not used further in the results or discussion. They only complicate the text flow. See some examples below.

Thank you for taking the time to read our manuscript multiple times and give us this thoughtful response. We have given our methods a major overhaul in response to the comments of this reviewer.

I would try to unify some terminology. Novel species is for me, newly introduced, and here in the text, it is distantly phylogenetically/functionally related. Alien is here focal species. Competitor is native species (it is strange to accept this point of view). Aliens are firstly divided into naturalized and invasive species; further, in the text, this point is not mentioned, and authors do not use the status of alien species for any explanation.

We have unified the terminology as requested. We use the words distinctiveness (rather than novelty), alien (we no long distinguish naturalized and invasive), and competition with the resident community (which is anything heterospecific to the focal plant—native and alien).

The authors work in a large group of habitats from forests to prairies. I wonder if there are no differences in competition rate of single species across habitats.

Unfortunately, we cannot test this idea. Most of our focal alien species do not live in multiple habitat types. Furthermore, we didn't conduct the experiment across habitat types for any of our focal alien species. We mention this in the discussion (421-424).

l. 88-89 I do not agree. Classification of alien status, e.g., causal/naturalized/invasive is not based on the relationship to the native species, but it is based on the distributional characteristic of the species

We can see how this was confusing. Because we never used the classification of alien status in any statistical text, we have streamlined the manuscript by removing this classification altogether.

l. 163-167 detail information, which is useless in the context of the study

We included this information because it was specifically asked for by the previous set of reviewers. However, now that we no longer have the classification of aliens in the manuscript, we have removed it.

l. 171-179 text describes focus on families Fabaceae and Brassicaceae in detail, but looking at table 1, the list of alien species is much broader. Further, the information about the families is not also used in the story. This paragraph could be deleted.

We have deleted this paragraph.

l. 198-201. It is not clear to me if the clipped biomass was somehow used further in the analyses.

We are now clear in the methods why this is an important variable and that we include it as a covariate in our analysis (L170-176, 263-266)

l. 245 phylogeny was pruned to include dicots only. I wonder if this is not a problem in plots from prairies and other non-forest vegetation types, where grasses built an essential part of the native communities. The phylogenetic distance is based on dicots only, and the value could be shifted based on occurrences of monocots.

We have re-done the phylogenetic analyses to include monocots at the genus level. We did not differentiate between various monocot species in the field (e.g. multiple Carex species a single site), and most sites had a single dominant genus (e.g. Andropogon species at our prairie sites).

l. 269-227 it is nice that the authors used several phylotrees to be sure that the results are not affected by the phylogeny. However, such information could be given in the supplementary.

This was included to address a comment made by a previous reviewer. We have moved it to the supplementary as requested.

l. 282-284 some of the named traits are further in the text mentioned as too correlated and not used for analyses. Can it be possible to add here only relevant traits?

We have moved this paragraph to Appendix 3, which describes our trait selection procedure in greater detail.

l. 282 wet/dry ratio - do you mean LDMC?

No. We mean wet:dry. We could calculate the inversion of this (LDMC), but we have not done this, and the correlation would still exist if we did.

l. 291 I wonder if the traits were measured for dicots only and if not - Am I right that different species were used for phylogenetic part of analyses and different for functional part? Were the same species used for both the parts of the study?

We only measured traits on dicots. We now state this directly in the methods

l. 371-372 probably should be given in the methods.

We have moved this to the methods as suggested.

Referee: 5

In the paper entitled 'Phylogenetic and functional novelty explain alien plant population responses to competition' the authors aim to find support for darwin's naturalization hypothesis, i.e. that evolutionary distinct alien plants should compete less with the resident flora. To do this, the authors use a removal experiment as well as parametrize matrix population and integral projection models, along with phylogenetic and trait information. They found that alien species that were more distinct to their competitors were indeed less affected by competition, although this pattern was not maintained when calculating relatedness at the regional level. They also found that the models explaining competition effects were improved when including functional distinctiveness. The authors argue that their findings support the naturalization hypothesis and they highlight both the scale and the functional vs phylogenetic information issues.

Generally, this paper was very hard to follow and needs much improvement in terms of structure and delivery of the message. I do think that the results are interesting and experimental evidence is surely needed, but in its present form the paper is not suited for publication. I have three major concerns which I explain bellow.

First, the authors fail to clearly state their aims and hypothesis (e.g. paragraphs at the end of the introduction in LL 134-148 describe methods and results). As a consequence, the whole text lacks a structure, with concepts presented in a random manner, and it would need major editing. The authors seemed to have slightly ameliorated the manuscript from the first revision. However, even though I did not perform the first revision, my opinion is that these changes were mostly superficial. What are exactly the questions that the authors are asking? What are the expectations regarding those questions?

It is true that our changes in the previous revision, we focused on the analysis comments and made only superficial changes to the introduction. We thought that this was appropriate at the time. However, this comment, combined with the comment of Referee #3, made it really clear to us that our introduction needed a major overhaul. We were not clear in the questions we are asking, and our structure didn't clearly set up the conceptual framework for those questions.

We have set up the following questions in our introduction:

We parameterize matrix projection and integral projection models for alien plant species in the presence and absence of competitors to ask whether phylogenetic distinctiveness predicts the strength of competitive interactions. We test whether the result found at a small spatial grain is robust even if the resident community is defined at a larger spatial grain that would be typical for studies using species checklists. Additionally, we quantify functional traits for 116 plant species and ask whether simultaneously incorporating information on functional traits and phylogenetic relationships

improves the relationship between distinctiveness and the strength of competitive interactions, and, if so, which functional traits play key roles in explaining the effect size of competition.

In addition, we have removed Darwin's naturalization conundrum as our conceptual framework, as this does not set up these questions. We have a new framing that we believe works much better (please see responses to Referee #3)

Second, it is unclear what the authors mean by 'alien success'. This derives from the fact that they do not explain early in the text the exact invasive stage at which they are working. They do mention several times that they work on the post-establishment phase, but I) they seem to want to test the Darwin's naturalization hypothesis, which as the name suggests, deals with naturalization, II) their definition of naturalized 'currently considered relatively benign' potentially includes also casual species (that are not able to maintain viable populations and therefore are not considered established), III) they discuss propagule pressure and environmental filtering, which are mechanisms known to influence the establishment phase rather than the spreading phase. The mechanism they directly assess, competition, is present at both stages. What invasion stage are the authors studying? What are their expectations regarding the role/importance of competition in that particular stage?

We really appreciated this comment, which made it clear to us that we needed to change the conceptual framing of our study. It is true, our work is neither about the naturalization stage of invasion nor is it about environmental filtering. We are clear that we are examining performance of already established aliens, and testing a class of hypothesis about the effects of competitors on the performance of alien plants.

Finally, I found the methods rather complex and not explained sufficiently. It is rather difficult for the reader to understand what was done and why. For example, it was not clear to me what was done exactly with the trait information and to what purpose, nor why the comparison between the functional and phylogenetic information was needed. The models seem far too complex (or maybe not explained clearly). Why not use a single full model with all functional and phylogenetic metrics, select the best model, and discuss the variables maintained?

We have revised many of our methods paragraphs to include a first sentence that explains why each analysis was performed. We have streamlined and reorganized the methods to make them easier to follow and less complex.

We do see the point about the utility of a single full model. However, the response variables in these models is the effect size of competition, for which we have 14 observations. We could not reasonably fit a model with 7 traits, a phylogenetic distance metric, and the competitor biomass as separate predictors. This necessitated some kind of dimensionality reduction of our predictors. Furthermore, the process of maximizing explained variance tested, and rejected, multiple combinations of traits, including the full set of traits. We have updated our methods to provide more details on the procedure and the discussion to include comments on the traits it retained.

Appendix C

Dear Professor Hans Heesterbeek,

We are very excited to publish this manuscript in *Proceedings B*, and are especially grateful for the additional round of revisions that you granted us. We have revised our manuscript to include the comments from the reviewer, and provide responses in italics below.

Best regards,

Sam Levin (on behalf of all co-authors)

Reviewer 3:

I encourage the authors to discuss why the relationship between distinctiveness and the effect size of competition disappeared at large scale. Blanchet et al 2020 Ecology Letters (10.1111/ele.13525) might help.

*Thank you for the suggestion. We have added a sentence that explicitly links our work to theirs (L381-383). Unfortunately, our experimental framework doesn't provide us with an opportunity to say **why** these effects aren't present, and we are cautious to expand upon this point much further as we were (rightfully) criticized for invoking too much about the potential for environmental filtering by a reviewer in the previous version of this manuscript.*

L56 use 'because' rather than 'due to' to keep the form parallel.

We have changed the wording.

L71 & L406-407 Here, the authors want to test why some aliens become invasive. But this requires to compare non-invasive aliens with invasive aliens. I would replace 'invasive' with 'successful' (or something similar) unless the authors include non-invasive vs invasive in the models. Besides, the authors have removed the categories of alien species (i.e. invasive vs non-invasive) in the current version. I would still keep them, otherwise some audience may think all species are invasive.

We have changed the wording.

L86 replace 'herbivore release' with 'herbivores'. If aliens are enemy released, there will be no indirect effects mediated by enemies.

We have changed the wording.

L87 To the best of my knowledge, Elton hadn't discussed (or even mentioned) Darwin's naturalization hypothesis in his 1958 book. I apologize if I am wrong.

We examined Elton's book in detail again, and you are correct. Elton discusses the relationship between diversity and invasibility, but doesn't specifically mention the connection between this idea and Darwin naturalization hypothesis. Thus, the way we

had referenced Elton was not correct. We have fixed this in the introduction by removing the Elton reference and focusing on Darwin's naturalization hypothesis.

L346 & 348 It will help to write the abbreviation in full form at the beginning of each section, as well as in the figure legends.

We have updated these to explicitly spell out the acronyms at the beginning of every section that uses them.

L418 What does 'invasion maturity' mean? Consider to reword.

We have changed the wording.